# Probabilistic size-and-shape functional mixed models

**Fangyi Wang**
Department of Statistics
The Ohio State University
Columbus, OH, 43210
wang.15022@osu.edu

**Karthik Bharath**
School of Mathematical Sciences
University of Nottingham
Nottingham, UK, NG7 2RD
Karthik.Bharath@nottingham.ac.uk

**Oksana Chkrebtii**
Department of Statistics
The Ohio State University
Columbus, OH, 43210
oksana@stat.osu.edu

**Sebastian Kurtek**
Department of Statistics
The Ohio State University
Columbus, OH, 43210
kurtek.1@stat.osu.edu

## Abstract

The reliable recovery and uncertainty quantification of a fixed effect function $\mu$ in a functional mixed model, for modelling population- and object-level variability in noisily observed functional data, is a notoriously challenging task: variations along the $x$ and $y$ axes are confounded with additive measurement error, and cannot in general be disentangled. The question then as to what properties of $\mu$ may be reliably recovered becomes important. We demonstrate that it is possible to recover the size-and-shape of a square-integrable $\mu$ under a Bayesian functional mixed model. The size-and-shape of $\mu$ is a geometric property invariant to a family of space-time unitary transformations, viewed as rotations of the Hilbert space, that jointly transform the $x$ and $y$ axes. A random object-level unitary transformation then captures size-and-shape *preserving* deviations of $\mu$ from an individual function, while a random linear term and measurement error capture size-and-shape *altering* deviations. The model is regularized by appropriate priors on the unitary transformations, posterior summaries of which may then be suitably interpreted as optimal data-driven rotations of a fixed orthonormal basis for the Hilbert space. Our numerical experiments demonstrate utility of the proposed model, and superiority over the current state-of-the-art.

## 1   Introduction

Consider a sample of functions from the much-studied Berkeley growth data in Figure 1(a), where growth rate curves from measurements on heights in centimeters of 54 girls and 39 boys from age 1 to 18 are plotted on a rescaled time axis. It appears that individuals experience different numbers of small and large growth spurts that differ in magnitude and timing. Any reasonable generative model for inference on distributional aspects of the functions will need to account for the fact that the sample size ($n = 93$) is smaller than the dimension of each observation, and it is thus common to consider tractable parametric probabilistic models. A popular choice is a functional mixed model

$$f_i = \mu + v_i + \epsilon_i, \quad i = 1, \dots, n, \tag{1}$$

comprising three component real-valued functions on $[0, 1]$: (i) fixed population-level function that represents an average, or representative, change in growth rate; (ii) an individual- or object-level

38th Conference on Neural Information Processing Systems (NeurIPS 2024).

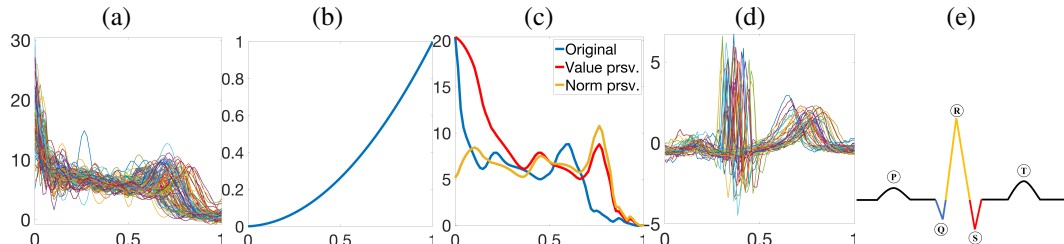

Figure 1: (a) Berkeley growth rate curves. (b) Convex phase function $\gamma$. (c) One example function $f$ from (a) (blue) transformed by value-preserving action $f \circ \gamma$ (red) and norm-preserving action $(f \circ \gamma)\sqrt{\dot{\gamma}}$ (yellow). Here, $f \circ \gamma$ has the same classical notion of shape as $f$, whereas $(f \circ \gamma)\sqrt{\dot{\gamma}}$ has the same size-and-shape as $f$ as described in Section 2. (d) PQRST complexes. (e) PQRST pattern: P wave (first max), QRS complex (sharp min-max-min) and T wave (last max) [Pham et al., 2023].

random function $v_i$ that represents deviations from $\mu$; and, (iii) a zero-mean random measurement error process $\epsilon_i$. The growth rate function $f_i(t)$ of individual $i$ at time $t$ differs pointwise from an average $\mu(t)$ via a smooth random translation $v_i(t)$ observed with additive error $\epsilon_i(t)$.

A function with any number of peaks representing spurts may be generated under the model, and as such exact recovery of $\mu$ is not possible without further constraints on $v_i$ and $\epsilon_i$. The issue is exacerbated when $f_i$ in (1) is changed to $f_i \circ \gamma_i$ by incorporating phase variability, or $x$-axis variability, via a time dilation/contraction object-level increasing function $\gamma_i : [0,1] \to [0,1]$, which models variability in timings of the spurts. Even for the simpler model, $f_i \circ \gamma_i = \sigma\mu + \epsilon_i$, for a fixed scalar $\sigma > 0$, recovery of $\mu$ is possible only when the measurement error process $\epsilon_i$ has a rank one covariance operator [Kurtek et al., 2011, Chakraborty and Panaretos, 2021].

The crux of the problem in reliable recovery of the population-level function $\mu$ lies in the fact that its topological features such as number of critical points and their nature (non-degenerate or otherwise), while preserved by $\gamma_i$, may be altered due to addition of $v_i$ and $\epsilon_i$. Consequently, a growth spurt (peak) in the average $\mu$ at time $t$ (or $\gamma_i(t)$) may either be preserved or destroyed. From a modeling perspective, uncertainty quantification of $\mu$ in (1) within a Bayesian framework is unreliable in the presence of phase $\gamma_i$, regardless of how informative the prior distributions on $v_i$ and $\gamma_i$ are. Figure 1(d)&(e) show an additional motivating dataset of PQRST complexes segmented from an electrocardiogram (ECG) signal and a typical PQRST pattern, respectively.

Summarily, under the functional mixed model (1), the topological shape of $\mu$ may be irrevocably altered, and thus ordinate values of $\mu$ may not be reliably inferred through probabilistic modeling. The goal of the paper is to investigate if the situation may be salvaged by considering a complementary geometric property of $\mu$, its size-and-shape, characterized by a joint transformation of its range and domain by the phase function $\gamma$.

**Contributions.** We focus on sampling from and summarizing the posterior distribution of a fixed effect function $\mu$ in a functional mixed model with random object-level phase and amplitude components, without a finite-rank covariance assumption on the error process. We do not make any claims on efficiency of the computational algorithms, but emphasize that the goal is to thoroughly investigate the novel size-and-shape perspective in inference on $\mu$. Thus, our contributions are as follows.

1. We propose a mixed model in a functional Hilbert space for the size-and-shape of a square-integrable fixed effect $\mu$ by considering an isometric action of the infinite-dimensional group of phase functions $\Gamma$. The geometric property of $\mu$ thus preserved is referred to as its size-and-shape (Section 2), which we demonstrate may be reliably inferred under the model. To our knowledge, this is the first paper to consider a Bayesian functional mixed effects model with an *unrestricted* form of phase variation, and employ the novel perspective of inferring the size-and-shape of $\mu$.

2. Informative priors on the phase functions $\gamma$ regularize the posterior of $\mu$, and sampling is assisted by exploiting the group structure of the phase functions while exploring the parameter space (Section 3 and Appendix D).

3. Isometric action of $\gamma$ engenders a unitary transformation of the Hilbert space; the class of unitary transformations indexed by $\gamma$ constitutes rotations of coordinates of the Hilbert space. Upon expressing $\mu$ and $v_i$ in a suitable orthonormal basis of the Hilbert space, inference for $\gamma_i$ translates to an automatic identification of a data-optimal rotation of the chosen basis that best captures population- and object-level variations, and performs better when compared to an empirical functional principal component analysis (FPCA) basis (Appendices B and G).

4. We carry out extensive numerical experiments to investigate utility of the proposed model, and demonstrate that the posterior mean of $\mu$ under our model better captures the properties of $\mu$ than the estimate given by current state-of-the-art [Claeskens et al., 2021](Section 4 and Appendix I).

**Related work.** In absence of phase functions as part of the random effect, approaches have broadly focused on two types of dimension reduction techniques for estimating the fixed effect: (i) empirical orthonormal basis from FPCA [e.g., Yao et al., 2005], and (ii) pre-specified basis functions [e.g., Rice and Wu, 2001, Guo, 2002, Chen and Wang, 2011, Zhu et al., 2011, Morris and Carroll, 2006, Huo et al., 2023]. Some recent works have extended methodology to multivariate functional mixed effect models [Volkmann et al., 2023], scalar on function regression [Liu et al., 2017], spatial-temporal variation [Zhu et al., 2019], and generalized functional mixed effect models [St. Ville et al., 2022].

In the presence of phase functions, the notable works are by Claeskens et al. [2021] and Raket et al. [2014], where the former provides sufficient conditions for exact recovery of the fixed effect when the error process is not assumed to have phase variation; in our numerical experiments, we compare our results to the estimator proposed by Claeskens et al. [2021], which represents the state-of-the-art.

We are unaware of work in literature on Bayesian approaches to functional mixed models with unrestricted, nonparametric phase functions to *jointly* model amplitude and phase variations, and not sequentially following pre-processing via registration. Of relevance, however, is the work by Schiratti et al. [2017], where manifold-valued curves with reparameterization variation are considered.

## 2 Phase functions and size-and-shape-preserving transformations

Without loss of generality, we assume that all functions are observed on a fixed domain $[0, 1]$. Let $\mathbb{L}^2([0, 1], \mathbb{R}) = \{f : [0, 1] \to \mathbb{R} \mid \int_0^1 |f(t)|^2 \mathrm{d}t < \infty\}$ (henceforth simply referred to as $\mathbb{L}^2$) denote the representation space of interest, i.e., the space of real-valued square-integrable functions on $[0, 1]$. The space $\mathbb{L}^2$ when endowed with the norm $\|f\| := [\int_0^1 |f(t)|^2 \mathrm{d}t]^{1/2}$ coming from the inner product $\langle f, g \rangle := \int_0^1 f(t)g(t)\mathrm{d}t$ is a Hilbert space. Additionally, let $\Gamma = \{\gamma : [0, 1] \to [0, 1] \mid \gamma(0) = 0, \gamma(1) = 1, \dot{\gamma} > 0\}$ denote the group of orientation-preserving diffeomorphisms ($\dot{\gamma}$ is the time derivative of $\gamma$) of $[0, 1]$. In the functional data analysis literature, $\Gamma$ is used to model phase variability in functional observations. Thus, elements of $\Gamma$ will henceforth be referred to as phase functions. Note that $\Gamma$ is a Lie group with composition $(\gamma_1, \gamma_2) \mapsto \gamma_1 \circ \gamma_2$ as the group operation, where $\circ$ is function composition. This implies that (i) $\Gamma$ is an infinite-dimensional smooth manifold, (ii) $(\gamma_1 \circ \gamma_2) \circ \gamma_3 = \gamma_1 \circ (\gamma_2 \circ \gamma_3)$ for any $\gamma_1, \gamma_2, \gamma_3 \in \Gamma$, (iii) it contains the identity element $\gamma_{id}(t) = t$ such that $\gamma \circ \gamma_{id} = \gamma$ for any $\gamma \in \Gamma$, and (iv) for any $\gamma \in \Gamma$ there exists $\gamma^{-1} \in \Gamma$ such that $\gamma \circ \gamma^{-1} = \gamma_{id}$. The group structure of $\Gamma$ plays a pivotal role when defining prior and proposal distributions for phase functions; this is further elucidated in Section 3.2 and Appendix D. Importantly, the group $\Gamma$ can act on the function space $\mathbb{L}^2$ from the right, engendering maps $\mathbb{L}^2 \times \Gamma \to \mathbb{L}^2$, in different ways, thus resulting in different notions of phase variation in functional data.

1. **Value-preserving action.** The value-preserving mapping is defined via composition: $f \circ \gamma$, $f \in \mathbb{L}^2$, $\gamma \in \Gamma$. This action is referred to as the time warping of a function since $f$ and $f \circ \gamma$ traverse the same exact $y$ axis values, but at different times ($x$ axis values). It is commonly used for alignment or registration of prominent features in functional data, e.g., local extrema, a process traditionally referred to as amplitude-phase separation.

2. **Area-preserving action.** The area-preserving mapping is defined as $(f \circ \gamma)\dot{\gamma}$, $f \in \mathbb{L}^2$, $\gamma \in \Gamma$. This action is commonly used for statistical analysis of probability density functions.

3. **Norm-preserving action.** The norm-preserving mapping is defined as $(f, \gamma) := (f \circ \gamma)\sqrt{\dot{\gamma}}$, $f \in \mathbb{L}^2$, $\gamma \in \Gamma$. An important property of this action is that it preserves the $\mathbb{L}^2$ norm of a function: $\|f\| = \|(f, \gamma)\|$. The action has been profitably used in the problem of function alignment/registration, wherein a desideratum is a cost function invariant to a simultaneous action of the group $\Gamma$, or time warping [Srivastava and Klassen, 2016, Chapter 4].

The operator $D_\gamma : \mathbb{L}^2 \to \mathbb{L}^2$, $f \mapsto D_\gamma(f) := (f \circ \gamma)\sqrt{\dot{\gamma}}$ arising from the norm-preserving action for a fixed $\gamma \in \Gamma$ is a surjective isometry, and hence unitary. Thus, $D_\gamma$, for a fixed $\gamma \in \Gamma$, is an infinite-dimensional rotation in the Hilbert space $\mathbb{L}^2$, and the group $\mathbb{D} := \{D_\gamma, \gamma \in \Gamma\}$ of rotations of $\mathbb{L}^2$ plays a prominent role in classical white noise calculus [Hida, 2015].

A general space-time diffeomorphism $(t, x) \mapsto (\sigma_1(t, x), \sigma_2(t, x))$ preserves the topology of the graph $\{(t, f(t))\}$ of $f$ viewed as a subset of $[0, 1] \times \mathbb{R}$. The operator $D_\gamma$ is associated with a special space-time diffeomorphism with $\sigma_1(t, x) = \gamma(t)$ and $\sigma_2(t, x) = x\sqrt{\dot{\gamma}(t)}$. In this sense, the size and shape of $f$, as it relates to its norm and its graph, is preserved by operators in $\mathbb{D}$.

To better understand how $D_\gamma$ transforms $f$, consider Figure 1(b)&(c). As seen with the red curve in (c), the value-preserving action of $\gamma$ preserves the image of $t \mapsto f(t)$, where ordinate values of $f$ are relabeled in an order-preserving way and no new values are created/destroyed. A more classical notion of the shape of $f$ is thus preserved under the value-preserving action in that the topology of the level sets of $f$ is unaltered. In contrast, the norm-preserving action shown with the yellow curve in (c) alters the image of $t \mapsto f(t)$, and may create new critical points, but in a way such that its size-and-shape in the sense alluded to above is preserved. As such, the norm-preserving action is size-and-shape preserving, i.e., $f$ and $D_\gamma(f)$ for any $\gamma \in \Gamma$ are equivalent in their size-and-shape.

Use of the operator $D_\gamma$ is particularly useful for modeling phase variation in model (1):

(i) Dimension reduction of the fixed and random effects is implemented via an orthonormal basis system, and operators in $\mathbb{D}$ allow us to rotate these basis systems to better align with observed data (see Appendix B). This provides modeling flexibility that is often needed due to the use of a finite, and potentially small, number of basis functions to represent these model components.

(ii) The norm preserving action can be viewed as a combination of value-preserving warping and an associated local scaling of function values, i.e., when $\dot{\gamma}(t) > 1$ ($\dot{\gamma}(t) < 1$) the function value $f(\gamma(t))$ is warped to the right (left) relative to $f(t)$ and additionally rescaled by a factor of $\sqrt{\dot{\gamma}(t)}$.

(iii) The equivalence class of functions having the same size-and-shape under the norm-preserving action is 'larger' than the one under the value-preserving action in the following sense. The map $\pi : \mathbb{L}^2 \to \mathbb{L}^2/\sim$ takes a function to its equivalence class determined by equivalence relation $\sim$. It can be shown that the measure of an equivalence class, under the pushforward of a non-degenerate measure with support in $\mathbb{L}^2$ under $\pi$, is larger when $\sim$ pertains to norm-preserving action as opposed to value-preserving one (see Example 6.5.2 in Bogachev [1998] for an example that illustrates this idea). In practice, the larger equivalence class under the norm-preserving action helps with reliable recovery of the size-and-shape of the fixed effect $\mu$, in contrast to the more traditional notion of shape of $\mu$ under the value-preserving action, wherein the additive linear term $v_i + \epsilon_i$ more easily alters the shape of $\mu$.

## 3 Bayesian size-and-shape functional mixed model

The operator $D_\gamma$ related to the norm-preserving action of $\Gamma$ is an isometry of $\mathbb{L}^2$. The use of Euclidean isometries arising from translations and rotations as size-and-shape preserving has a long history in statistical shape analysis of landmark configurations [e.g., Dryden and Mardia, 2016]. To elaborate, let $X_i \in \mathbb{R}^{K \times 2}$ denote a set of $K$ landmarks, i.e., $X_i$ is a matrix that contains the coordinates of $K$ points in $\mathbb{R}^2$, and $SO(2) = \{R \in \mathbb{R}^{2 \times 2} | R^T R = RR^T = I, \det(R) = 1\}$ denote the rotation group with matrix multiplication as the group operation. Consider the following perturbation (generative) model for the size-and-shape of an object represented via a landmark configuration $X_i$:

$$X_i = (M + E_i)R_i + \mathbb{1}t_i^T, \tag{2}$$

where $t_i \in \mathbb{R}^2$ is a random translation, $\mathbb{1} \in \mathbb{R}^K$ is a vector of 1s, $R_i \in SO(2)$ is a random rotation, $M \in \mathbb{R}^{K \times 2}$ is a fixed template, and $E_i \in \mathbb{R}^{K \times 2}$ is the error. Using this model, it is of primary interest to estimate the fixed effect (size-and-shape) $M$ based on observed landmark configurations $X_1, \ldots, X_n$. Transformation of the model components $M$ and $E_i$ using rotations $R_i$ (and translations) in order to align them to the observation $X_i$ may be viewed as an isometric, or norm-preserving, transformation of the coordinate system for $(M + E_i)$, so that *only the size-and-shape of $M$ can be reliably recovered, but not $M$* [Lele and McCulloch, 2002].

Inspired by size-and-shape analysis of landmark data, and in contrast to existing works using the value-preserving action [Raket et al., 2014, Claeskens et al., 2021], our proposed functional mixed effects model utilizes the norm-preserving action of $\Gamma$ on $\mathbb{L}^2$. Analogous to the model in (2), for a function $f_i \in \mathbb{L}^2$, one can define a functional perturbation model $f_i = D_{\gamma_i}(\mu + \epsilon_i) = [(\mu + \epsilon_i) \circ \gamma_i]\sqrt{\dot{\gamma}_i}$ for a template function $\mu$ with random phase functions $\gamma_i$, wherein the $\mathbb{L}^2$ coordinate system of the model components $\mu$ and $\epsilon_i$ is aligned to the coordinate system of the function $f_i$ via a rotation using $\gamma_i$.

To the fixed effect, or signal plus noise model, one can additionally introduce an object-level random effect to increase modeling flexibility, resulting in:

$$f_i = D_{\gamma_i}(\mu + v_i + \epsilon_i) = [(\mu + v_i + \epsilon_i) \circ \gamma_i]\sqrt{\dot{\gamma}_i}. \tag{3}$$

Thus, $\gamma_i$ *and* $v_i$ *act as size-and-shape preserving and altering random effects, respectively*. As in the landmark case, the primary goal of interest is to estimate $\mu$ and quantify its uncertainty using independent observations $f_1, \ldots, f_n$, and it is possible to reliably infer only the size-and-shape of $\mu$.

While the model in (3) is written in terms of infinite-dimensional objects, functional data is often observed under a common discretization at time points $t_1 = 0 < t_2 < \cdots < t_{T-1} < t_T = 1$. The assumption of a common discretization for all functional observations is adopted for clarity and succinctness in presenting the proposed Bayesian model; it can be easily relaxed to accommodate more general scenarios. The model is quite flexible and more realistic assumptions (e.g., arbitrary error dependence structure such as Matérn covariance, or non-Gaussian) may easily be incorporated at some computational cost. We instead focus on the phase component, and demonstrate the utility of the size-and-shape perspective in inferring geometric properties of $\mu$.

Let $\boldsymbol{f}_i = (f_i(t_1), \ldots, f_i(t_T))^\top \in \mathbb{R}^T$, $i = 1, \ldots, n$ represent the discretized function values for each observation. The discretized observation model is then given by

$$f_i(t_j) = [(\mu + v_i + \epsilon_i) \circ \gamma_i](t_j)\sqrt{\dot{\gamma}_i(t_j)}, \ i = 1, \ldots, n, \ j = 1, \ldots, T. \tag{4}$$

Letting $\{\phi_k\}_{k=1}^\infty$ and $\{\tilde{\phi}_k\}_{k=1}^\infty$ denote two orthonormal basis systems for $\mathbb{L}^2$, we represent the model components $\mu$ and $v_i$ as linear combinations of basis functions. For dimension reduction, we define $\mu := \sum_{k=1}^{B_f} a_k \phi_k$ and $v_i := \sum_{k=1}^{B_r} c_{i,k} \tilde{\phi}_k$, i.e., the two basis sets are truncated to $B_f$ and $B_r$ basis functions for the fixed and (size-and-shape altering) random effects, respectively. We assume that the random effect coefficients are independent and identically normally distributed (iid) $c_{i,k} \sim N(0, \sigma_c^2)$. The discretized error is assumed to be iid $\epsilon_i(t_j) \sim N(0, \sigma^2)$.

### 3.1 Choice of basis functions and likelihood

Specifications of $\mu$ and $v_i$ require appropriate orthonormal basis functions, and we consider modified Fourier basis or the B-spline basis for $\mu$, and the B-spline basis for $v_i$. The modified Fourier basis contains the following elements $\{\sqrt{3}t, \ \sqrt{3}(1-t), \ \sqrt{2}\cos(2\pi jt), \ \sqrt{2}\sin(2\pi jt); \ j = 1, \ 2, \ldots; \ t \in [0, 1]\}$ and is subsequently orthonormalized via the Gram-Schmidt procedure under the $\mathbb{L}^2$ metric; larger values of $j$ yield basis functions with finer harmonics over the domain $[0, 1]$. This basis is used to specify the fixed effect function $\mu$ only as it is effective at representing global periodic trends and oscillations. On the other hand, the B-spline basis is defined locally via piecewise polynomials, and is therefore better at capturing local variation and finer function features; we also orthonormalize the B-spline basis via the Gram-Schmidt procedure under the $\mathbb{L}^2$ metric. In some cases where the underlying $\mu$ is expected to exhibit more local features, we use the B-spline basis rather than the modified Fourier basis. Finally, one needs to choose the number of basis functions to model $\mu$ and $v_i$. This choice depends on the application of interest. For example, one may use a larger number of basis functions for $\mu$ when the observations have a complex shared global structure. Effects of misspecifying the number of basis functions for $\mu$ and $v_i$ are studied in Appendix H.

The main inferential tasks of interest are to estimate and assess uncertainty in (i) mean size-and-shape $\mu$ via the coefficient vector $\boldsymbol{a} = (a_1, \ldots, a_{B_f})^\top$, (ii) variance of the size-and-shape altering random effect coefficients $\sigma_c^2$, and (iii) error variance $\sigma^2$. Thus, to simplify the inference, we marginalize the likelihood with respect to the size-and-shape altering random effect. Let $\Phi_i \in \mathbb{R}^{T \times B_f}$ denote the matrix of evaluations of the fixed effect basis, whose $(j, k)$th entry is given by $(\phi_k \circ \gamma_i)(t_j)\sqrt{\dot{\gamma}_i(t_j)}$. One can view $\Phi_i$ as a discretization of the rotated coordinate system, via the norm-preserving action that uses $\gamma_i$, for $\mu$. Similarly, let $\tilde{\Phi}_i \in \mathbb{R}^{T \times B_r}$ be the matrix of evaluations of the random effect basis, whose $(j, k)$th entry is $(\tilde{\phi}_k \circ \gamma_i)(t_j)\sqrt{\dot{\gamma}_i(t_j)}$. Let $\boldsymbol{a} \in \mathbb{R}^{B_f}$ and $\boldsymbol{c}_i \in \mathbb{R}^{B_r}$, $i = 1, \ldots, n$ be the vectors of basis coefficients for $\mu$ and $v_i$, $i = 1, \ldots, n$, respectively. Finally, let $\boldsymbol{\epsilon}_i^{\gamma_i} \in \mathbb{R}^T$, $i = 1, \ldots, n$ be the vectors of $\gamma_i$-transformed observation errors, with entries $(\epsilon_i \circ \gamma_i)(t_j)\sqrt{\dot{\gamma}_i(t_j)}$, $j = 1, \ldots, T$ that are independently distributed as $N(0, \sigma^2\dot{\gamma}_i(t_j))$, conditional on $\gamma_i$; the additional time-dependent scaling of the error variance comes from the norm-preserving action. Using this simplified notation, we can rewrite the model in (4) in matrix form as

$$\boldsymbol{f}_i = \Phi_i\boldsymbol{a} + \tilde{\Phi}_i\boldsymbol{c}_i + \boldsymbol{\epsilon}_i^{\gamma_i}, \ i = 1, \ldots, n, \tag{5}$$

which may be usefully compared to the size-and-shape perturbation model (2) for landmark configurations. With MVN used to denote the multivariate normal distribution, the distribution of $\boldsymbol{f}_i$, conditional on the vector of coefficients $\boldsymbol{c}_i$, is $\boldsymbol{f}_i | \boldsymbol{c}_i \sim \text{MVN}(\Phi_i \boldsymbol{a} + \tilde{\Phi}_i \boldsymbol{c}_i, \sigma^2 \text{diag}(\dot{\gamma}_i(t_j)))$, where $\text{diag}(\dot{\gamma}_i(t_j))$ is a $T \times T$ diagonal matrix whose $j$th diagonal element is $\dot{\gamma}_i(t_j)$. Then, the following marginal distribution of $\boldsymbol{f}_i$ is used to define the likelihood function (see Appendix C for full derivation):

$$\boldsymbol{f}_i \sim \text{MVN}(\Phi_i \boldsymbol{a}, \sigma^2 \text{diag}(\dot{\gamma}_i(t_j)) + \sigma_c^2 \tilde{\Phi}_i \tilde{\Phi}_i^T). \tag{6}$$

## 3.2 Prior distributions

The model parameters that need to be estimated are the fixed effect coefficients $\boldsymbol{a}$, random effect variance $\sigma_c^2$, variance of observation error $\sigma^2$, and individual phase functions $\gamma_i$. For $\boldsymbol{a}$, $\sigma_c^2$ and $\sigma^2$, we use weakly informative prior distributions: $\boldsymbol{a} \sim \text{MVN}(\boldsymbol{0}, 10000 \mathbf{I}_{\mathbf{B_f}})$, $\sigma_c^2 \sim \text{IG}(0.01, 0.01)$, $\sigma^2 \sim \text{IG}(0.01, 0.01)$ (IG is the inverse-gamma distribution).

The prior distribution over the space of phase functions $\Gamma$, to model the shape-and-size preserving random effect, can be specified in different ways [e.g., Telesca and Inoue, 2008, Bigot, 2013, Lu et al., 2017]. In this work, we use two tractable prior distribution models on $\Gamma$ to guard against confounding of inference between $\{\gamma_i\}$ and $\mu$: (i) a one-parameter family; (ii) a nonparametric finite-dimensional family compatible with the time discretization. Importantly, the prior models rely on the group structure of $\Gamma$.

- **Prior Model 1 (PM1) on $\Gamma$.** Each phase function $\gamma_i$ is defined via a single parameter $\alpha_i$ (Section 5.2 in Srivastava and Klassen [2016]) as

$$\gamma_i(t) = t + \alpha_i t(t-1), \ \alpha_i \in (-1, 1), \ t \in [0, 1]. \tag{7}$$

  The phase functions defined in (7) form a one-dimensional subset of $\Gamma$, making posterior inference more tractable with significant reductions in computational complexity. This subset contains phase functions with $\dot{\gamma}(t) \in (0, 2)$ for all $t \in (0, 1)$ and no inflection points. At the same time, we sacrifice flexibility in terms of the allowed rotations of $\mathbb{L}^2$. The $\alpha_i$, $i = 1, \ldots, n$ are assumed to be independent *a priori* following the $\text{Uniform}(-1, 1)$ distribution.

- **Prior Model 2 (PM2) on $\Gamma$.** A more flexible nonparametric prior model utilizes a point process-based prior distribution related to the Dirichlet process on $\Gamma$ [Bharath and Kurtek, 2020]. Under discretized time, each phase function may be represented by a finite number of successive increments $p(\gamma_i) = (\gamma_i(t_2) - \gamma_i(0), \ldots, \gamma_i(t_j) - \gamma_i(t_{j-1}), \ldots, \gamma_i(1) - \gamma_i(t_{T_\gamma - 1})) \in \mathbb{R}^{T_\gamma - 1}$; as $T_\gamma \to \infty$, the resulting prior has dense support in $\Gamma$. Then, the finite-dimensional prior distribution is placed on the vector of phase increments,

$$p(\gamma_i) \overset{ind}{\sim} \text{Dirichlet}(\theta_\gamma \boldsymbol{t}), \tag{8}$$

  where $\boldsymbol{t} = (t_2, t_3 - t_2, \ldots, 1 - t_{T_\gamma - 1})$ is defined via $T_\gamma$ consecutive time points on $[0, 1]$ and $\theta_\gamma$ is a precision parameter. First, the time points defining $\boldsymbol{t}$ can be different from the time points at which the functional observations were recorded. In fact, we usually choose $T_\gamma$ to be relatively small, e.g., five or seven, to simplify the prior model. Second, a large value for $\theta_\gamma$ regularizes the phase functions toward $\gamma_{id}$. We use $\theta_\gamma = 30$ in all of our numerical experiments. The resulting $\gamma_i$ is a piecewise linear function with changes in the slope $\dot{\gamma}_i$ at $t_2, \ldots, t_{T_\gamma - 1}$. This prior distribution is more flexible than the one in PM1. However, depending on the chosen number of discretization points $T_\gamma$, the dimension of the phase parameter space can be much larger in this case, making posterior inference more challenging. Interested readers can refer to Matuk et al. [2022] for more details behind this choice of prior distribution on phase functions in the context of Bayesian modeling for functional data.

Posterior inference on all parameters is conducted via Markov chain Monte Carlo (MCMC) sampling using the Metropolis-Hastings algorithm. To efficiently explore the parameter space, we use adaptive proposal distributions. We monitor MCMC convergence using standard diagnostic plots, e.g., trace plots and autocorrelation plots. Trace plots provided in Appendix F indicate good convergence for all of the examples presented in this manuscript. Full details of the proposal distributions and the MCMC algorithm are in Appendices D and E.

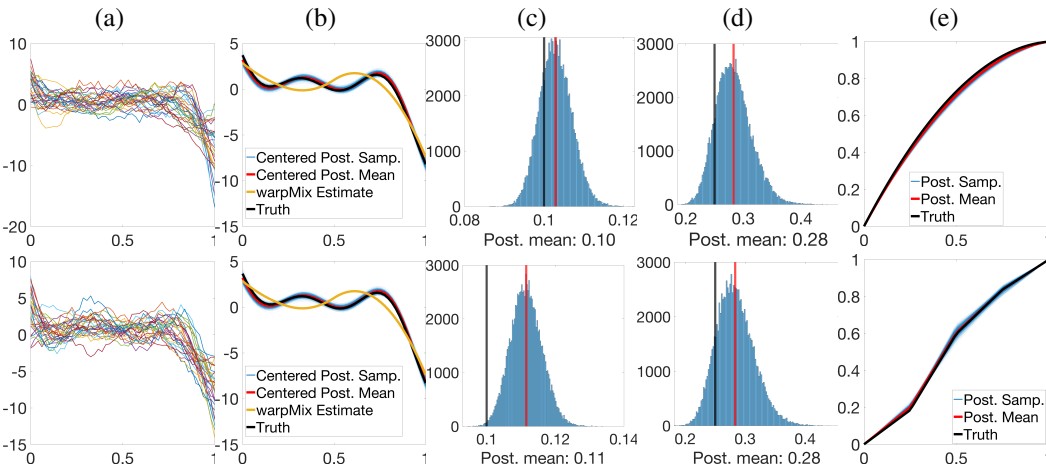

Figure 2: Row 1: Phase functions from PM1. Row 2: Phase functions from PM2. (a) Simulated data ($n = 30$). (b) Estimation of $\mu$: ground truth (black), posterior samples (blue), posterior mean (red), `warpMix` esimate (yellow). (c)&(d) Histograms of posterior samples for $\sigma^2$ and $\sigma_c^2$, respectively (posterior mean in red; ground truth in black). (e) Estimation of phase function for a randomly chosen observation: ground truth (black), posterior samples (blue), posterior mean (red).

## 4 Numerical experiments

We present posterior inference results from the model described in Section 3 for simulated and real data. Throughout, we compare our results to those generated by `warpMix`, a state-of-the-art frequentist functional mixed model [Claeskens et al., 2021]; we use the default parameter settings in `warpMix`. In addition, in Appendix A, we compare our results to those generated by a state-of-the-art Bayesian approach [Cheng et al., 2016], which models functions under the popular square-root velocity transformation [Srivastava et al., 2011b] as realizations of a Gaussian process centered at a mean function $\mu$. Phase variation is incorporated via the value-preserving action of $\Gamma$ with a prior model that is the same as PM2 in our model.

The main object of interest for posterior inference is $\mu$. Note however that, as discussed earlier, $\mu$ and $D_\gamma(\mu) = (\mu \circ \gamma)\sqrt{\dot{\gamma}}$ for any $\gamma \in \Gamma$ are equivalent in terms of their size-and-shape. *Thus, we first center all posterior samples for $\mu$, via an average of the estimated posterior means of object-level phase functions, to obtain size-and-shape representatives in their equivalence classes under the norm-preserving action*; this centering is similar in spirit to the orbit centering in Srivastava et al. [2011b]. Assuming that we have $N$ posterior samples of each $\gamma_i$, $i = 1, \ldots, n$, we first compute $\bar{\gamma} := 1/(nN) \sum_{i=1}^n \sum_{j=1}^N \hat{\gamma}_i^j$, where $\hat{\gamma}_i^j$ is the $j$th posterior sample of the phase function $\gamma_i$; due to convexity of $\Gamma$, note that $\bar{\gamma} \in \Gamma$. We then compute the centered posterior samples of $\mu$, $(\hat{\mu}^j \circ \bar{\gamma})\sqrt{\dot{\bar{\gamma}}}$, which may be visualized directly or used to estimate posterior summaries, e.g., pointwise posterior mean and credible interval. In some cases, we also visualize the posterior samples and pointwise posterior mean for a phase function $\gamma_i$. For all examples in this section, we use $N = 100,000$ with a burn-in period of $200,000$ iterations; for visualization of posterior samples for $\mu$ and $\gamma_i$, we use a uniform subsample of size $1,000$ to ensure that all plots are easily readable.

### 4.1 Simulations

**Example 1: data generated from our model.** We first consider an example based on data simulated from model in (6). We use $B_f = 6$ modified Fourier basis functions for $\mu$ and $B_r = 6$ B-spline basis functions for each $v_i$. The ground truth variances are $\sigma_c^2 = 0.25$ and $\sigma^2 = 0.1$. Then, to generate the data, we sample $\boldsymbol{a} \sim \text{MVN}(\boldsymbol{0}, \mathbf{I_{B_f}})$, and consider two cases for phase functions based on PM1 and PM2: (i) $\alpha_i \overset{iid}{\sim} \text{Uniform}(-1, 1)$, and (2) $p(\gamma_i) \overset{iid}{\sim} \text{Dirichlet}(30\boldsymbol{t})$, $\boldsymbol{t} = (0, 0.25, 0.5, 0.75, 1)$. The sample size in this simulated example is $n = 30$.

Figure 2 displays results based on data generated via PM1 for phase functions (row 1) and PM2 (row 2). The simulated data is shown in panel (a). Upon visual inspection, it is difficult to discern the underlying $\mu$. Panel (b) displays estimation results for $\mu$: ground truth in black, centered posterior samples in blue, centered posterior mean in red, and `warpMix` estimate in yellow. The proposed

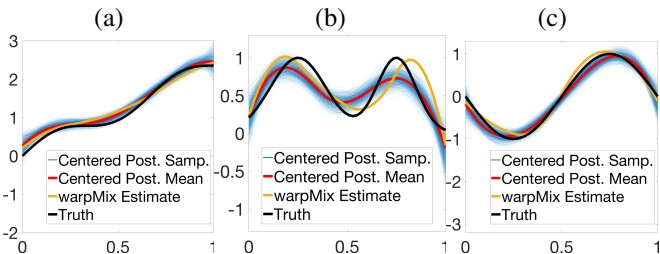

Figure 3: Comparison of estimation results based on Model 2-B and `warpMix` for (a) $\mu_1$, (b) $\mu_2$ and (c) $\mu_3$. In each panel, we show the ground truth (black), centered posterior samples (blue), centered posterior mean (red), and `warpMix` estimate (yellow).

Table 1: Comparison of fixed effect estimation accuracy based on posterior mean from proposed Bayesian models and `warpMix` estimate. Smallest estimation errors are highlighted in bold.

|         | `warpMix` | Model 1-F | Model 1-B | Model 2-F | Model 2-B |
|---------|-----------|-----------|-----------|-----------|-----------|
| $\mu_1$ | 0.0179    | 0.0194    | **0.0151**| 0.0193    | 0.0182    |
| $\mu_2$ | 0.0394    | 0.0134    | 0.0235    | **0.0070**| 0.0152    |
| $\mu_3$ | 0.0077    | 0.0195    | 0.0111    | 0.0044    | **0.0033**|

model is effective at recovering the underlying size-and-shape of $\mu$, which contains two local minima and maxima. On the other hand, the `warpMix` estimate only contains one local minimum and maximum, and is clearly an inaccurate representation of the size-and-shape of $\mu$. Panels (c) and (d) show histograms of posterior samples for the variance parameters $\sigma^2$ and $\sigma_c^2$, respectively; the ground truth values are in black and the estimated posterior means in red. In both cases, we slightly overestimate both variance parameters. Finally, in (e), we show estimation results for a phase function $\gamma_i$ corresponding to a randomly chosen observation. As before, posterior samples are shown in blue, posterior mean in red and the ground truth in black. In both cases (phase function generated from PM1 or PM2), we reliably recover the underlying ground truth.

**Example 2: data generated from `warpMix` model.** Next, we consider a more challenging scenario for the proposed model and specifically focus on a comparison to `warpMix`. In this case, phase functions and random effect functions are generated using the `warpMix` model; we set $\sigma_c^2 = 0.25$ and $\sigma^2 = 0.0001$. Comparison of estimation accuracy is based on three different fixed effect functions: $\mu_1(t) = \{\sin(3\pi t) + 3\pi t\}/4$, $\mu_2(t) = \exp^{-(t-0.25)^2/0.04} + \exp^{-(t-0.75)^2/0.02}$ and $\mu_3(t) = \cos(2\pi t + \pi/2)$, $t \in [0, 1]$. To generate the data, we use the value-preserving action as in the `warpMix` specification, and not the norm-preserving action that is utilized in our model.

To specify our models, we use (i) $B_f = 6$ modified Fourier basis functions or B-spline basis functions for $\mu$ with PM1 or PM2 for phase functions with $T_\gamma = 7$, and (ii) $B_r = 6$ B-spline basis functions for each $v_i$. In total, we consider four Bayesian models, 1-F, 1-B, 2-F and 2-B, where the number indexes the prior model on phase and the letter indexes the basis used to model $\mu$.

First, in Figure 3, we present estimation results for (a) $\mu_1$, (b) $\mu_2$ and (c) $\mu_3$ based on Model 2-B. The ground truth is in black, centered posterior samples in blue, centered posterior mean in red, and `warpMix` estimate in yellow. Visually, both `warpMix` and Model 2-B are effective at recovering the underlying fixed effect functions. To quantitatively assess estimation accuracy, we adopt the evaluation criterion used in Claeskens et al. [2021]: $\Delta_\mu := \sum_{j=1}^{T-1} [\hat\mu(t_j) - \mu(t_j)]^2 (t_{j+1} - t_j)$. The $\hat\mu$ for our models is defined as the centered posterior mean. Table 1 reports the results with best performance highlighted in bold. In all three cases, one of our Bayesian models yields the lowest estimation error. For $\mu_1$, only Model 1-B outperforms `warpMix`, which is not surprising since $\mu_1$ has the simplest structure. Nonetheless, the other three Bayesian models are competitive as well. For $\mu_2$, all four of our models significantly outperform `warpMix`, with Model 2-F yielding an error reduction of 82% as compared to `warpMix`. For $\mu_3$, the two models with PM2 on $\Gamma$ outperform `warpMix` suggesting the need for flexible phase functions in this case. These results show that the proposed models are effective in estimating $\mu$ even when the underlying data generating process uses value-preserving warping, and supports claim (iii) at the end of Section 2.

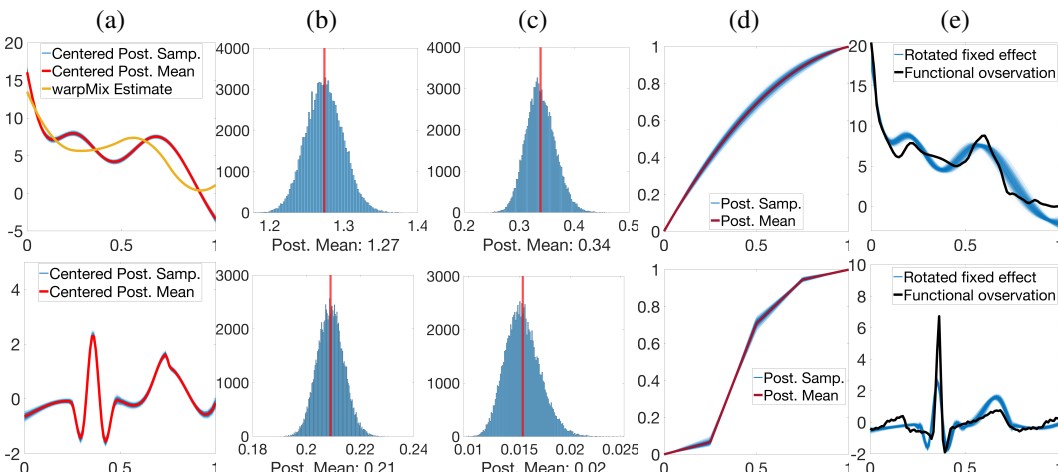

Figure 4: Estimation results for Berkeley data (row 1) and PQRST complexes (row 2). (a) Posterior samples (blue) and posterior mean (red) of $\mu$, and `warpMix` estimate (yellow). The `warpMix` model was unable to yield an estimate of $\mu$ for PQRST data. (b)&(c) Histograms of posterior samples for $\sigma^2$ and $\sigma_c^2$, respectively (posterior mean in red). (d) Posterior samples (blue) and mean (red) of phase function for a randomly chosen observation. (e) Observation corresponding to (d) (black) with rotated posterior samples of $\mu$ (blue).

## 4.2 Real data examples

We now consider application of the proposed modeling framework to (i) Berkeley growth rate functions ($n = 93$) [Srivastava et al., 2011b] (Figure 1(a)), and (ii) PQRST complexes ($n = 40$) [Kurtek et al., 2013] (Figure 1(d)). Our primary interest in the Berkeley data lies in estimating an average pattern of growth spurts. However, the number and magnitudes of growth spurts for each child may differ quite markedly from those in the average growth rate function. Majority of the PQRST complexes exhibit similar pattern of local extrema and we are interested in inferring the average pattern, while accounting for variability in magnitudes of the extrema across observations. A functional mixed effects model is thus appropriate for both these data settings to reliably estimate the size-and-shape of $\mu$.

For **Berkeley** data, we use modified Fourier basis for $\mu$ with $B_f = 6$, B-spline basis for each $v_i$ with $B_r = 6$, and PM1 on $\Gamma$; for **PQRST** data we use B-spline basis for $\mu$ with $B_f = 12$, B-spline basis for each $v_i$ with $B_r = 6$ to better model the sharp local features of the QRS complex (Figure 1(e)), and the PM2 model on $\Gamma$ to allow for inflection points in estimated phase.

Row 1 in Figure 4 shows estimation results for the Berkeley data. In (a), we show centered posterior samples of $\mu$ in blue with the centered posterior mean in red. We uncover two growth spurts, a small initial one followed by a larger pubertal one. This agrees with previous literature that has considered this data [Srivastava et al., 2011b]. The marginal posterior uncertainty for $\mu$ is very small throughout the domain. We also show the `warpMix` estimate, which was only able to recover one small growth spurt. Panels (b) and (c) show histograms of posterior samples of $\sigma^2$ and $\sigma_c^2$, respectively. Panel (d) displays the posterior samples (blue) and posterior mean (red) of a phase function for a randomly chosen observation. Panel (e) shows the observation corresponding to panel (d) in black. In addition, the blue functions are $D_{\hat{\gamma}_i^j}(\hat{\mu}^j)$, the posterior samples of $\mu$ rotated using corresponding posterior samples of the phase function, where $i, j$ index the observation and posterior sample, respectively. Note that the blue functions fit the observed black function well.

Row 2 in Figure 4 shows estimation results for the PQRST data. Panels (a)-(e) are the same as in the previous description. The estimated size-and-shape of $\mu$ resembles a PQRST complex as desired and contains sharp features that are representative of the observed data. Further, posterior uncertainty appears greater along the P and T waves than the QRS complex. The `warpMix` model failed to yield an estimate in this case, potentially due to lack of modeling flexibility in capturing the QRS complex and phase variation. The estimated phase function in (d) contains an inflection point motivating our use of PM2 on $\Gamma$. Finally, panel (e) shows that the rotated posterior samples of $\mu$ fit a randomly chosen observation well. Appendix I shows posterior mean and $95\%$ credible interval estimates, as well as `warpMix` estimates, of $\mu$ for five additional datasets.

# 5 Discussion

Numerical experiments in Section 4 and Appendices A and I demonstrate benefits of the proposed mixed model in recovering the size-and-shape of a fixed effect function $\mu$, while outperforming current state-of-the-art. What is lacking is theoretical support for the same, and this is work in progress.

To evolve the MCMC algorithm in MATLAB R2021a for $300,000$ iterations yielding $100,000$ posterior samples after burn-in, on a computing server with 6 parallel Intel(R) Xeon(R) CPUs with 20GB of memory, the computing time is approximately 93 and 111 minutes, respectively, under PMs 1 and 2 on $\Gamma$, based on $n = 30$ functions discretized at $T = 50$ points. There is room for improvement of efficiency in the MCMC computations, and alternatives may be explored.

We specify the number of basis functions for the fixed effect $\mu$ and the size-and-shape altering random effect $v_i$ *a priori*. Alternatively, one could treat the number of basis functions $B_f$ and $B_r$ as random and estimate them. This, however, would require more advanced MCMC algorithms for posterior inference. Further, we use the modified Fourier basis and B-spline basis in our model. Other orthonormal basis functions could also be used, but this choice is not crucial for reliable recovery of the size-and-shape of $\mu$ since the norm preserving action $D_\gamma$ rotates the basis system toward the data, allowing us to learn a data-driven basis for $\mu$. An alternative approach would be to directly learn an appropriate subspace for $\mu$, which we plan to consider in future work.

Exciting and novel extensions of the proposed mixed modeling framework are readily available to handle more complex functional data. The proposed model may be easily modified to handle sparsely/irregularly sampled and fragmented, or partially observed, functional data [Matuk et al., 2022]. The norm-preserving action of $\Gamma$ may be used to perform inference for the size-and-shape of a fixed effect parameterized open/closed curve $\mu : [0,1] \to \mathbb{R}^d$, $d > 1$ [Srivastava et al., 2011a, Kurtek et al., 2012]. Finally, for two-dimensional parameterized surfaces $f : D \subset \mathbb{R}^2 \to \mathbb{R}^3$, a similar norm-preserving action of the reparameterization diffeomorphism group with elements $\gamma : D \to D$ [Jermyn et al., 2017] may be used to infer the mean size-and shape of a fixed effect surface.

Functional data is arising as a common object in various applications, including computer vision and biomedical imaging, and functional data models are becoming increasingly important in machine learning [Rao and Reimherr, 2023a,b]. The ideas we have explored in the proposed framework can be applied more broadly to other regimes. For instance, employing mixed models with random effects to better model correlated input data in neural networks is fast gaining traction [Simchoni and Rosset, 2023]. There is also increasing interest in the use of geometry and invariance to nuisance transformations in neural networks [Bronstein et al., 2021]. Our framework provides understanding for the type of signal that can be recovered from complex input data in the presence of nontrivial symmetries and geometric information, which offers a new perspective on incorporating geometric constraints in various types of data settings and models.

## Acknowledgments and Disclosure of Funding

This research was partially funded by NIH R37-CA214955 (to SK and KB), NSF DMS-2015374, EPSRC EP/V048104/1 (to KB), and NSF CCF-1740761 and NSF DMS-2015226 (to SK). The authors have no competing interests to disclose.

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

We include the following appendices as support for the content in the main paper:

A. Additional comparison of estimation accuracy with state-of-the-art methods.

B. Additional motivation behind the norm-preserving action of $\Gamma$ on $\mathbb{L}^2$.

C. Detailed derivation of marginal likelihood in (6).

D. Specifications of all proposal distributions and derivation of the Metropolis-Hastings ratio for all model parameters.

E. Detailed Markov chain Monte Carlo algorithm to sample from the posterior distribution.

F. MCMC diagnostic plots for Example 1 in Section 4.1 and the two real data examples in Section 4.2.

G. Results of using an empirical FPCA basis to model the fixed effect function $\mu$.

H. Results of sensitivity analyses to assess effects of hyperparameter misspecification.

I. Estimation results for five additional real data examples.

## A  Additional comparison of estimation accuracy

Table 2 reports a comprehensive quantitative evaluation, in terms of estimation accuracy for the fixed effect function $\mu$, on the five simulated datasets used in Section 4.1. Rows 1-2 consider data simulated from our model under Prior Models 1 (PM 1) and 2 (PM 2) for phase functions. Rows 3-5 consider data simulated using the `warpMix` model with default parameter values. We compare estimation results produced using our model (columns Model 1-F through Model 2-B, where the number indicates the prior model on phase and the letters F (Fourier) or B (B-spline) correspond to the type of basis used to model the fixed effect function $\mu$) to those produced using `warpMix` [Claeskens et al., 2021] and the Bayesian model proposed in Cheng et al. [2016] (BRFC). As seen in the table, our model outperforms `warpMix` and BRFC in all of these simulation scenarios.

Table 2: Comparison of fixed effect estimation accuracy based on posterior mean from proposed Bayesian models, and `BRFC` (posterior mean) and `warpMix` estimates. Model 1-F to Model 2-B correspond to our models where the number indicates the prior model on phase and the letter corresponds to the type of basis used to model the fixed effect function (F=Fourier, B=B-splines). Smallest errors are highlighted in bold.

|  | warpMix | BRFC | Model 1-F | Model 1-B | Model 2-F | Model 2-B |
|---|---|---|---|---|---|---|
| PM1-F | 0.7972 | 2.5738 | **0.0452** | 0.2542 | 0.0746 | 0.4039 |
| PM2-F | 0.6417 | 2.1379 | 0.1152 | 0.2457 | **0.0539** | 0.2878 |
| warpMix-$\mu_1$ | 0.0179 | 0.1627 | 0.0194 | **0.0151** | 0.0193 | 0.0182 |
| warpMix-$\mu_2$ | 0.0394 | 0.0103 | 0.0134 | 0.0235 | **0.0070** | 0.0152 |
| warpMix-$\mu_3$ | 0.0077 | 0.0582 | 0.0195 | 0.0111 | 0.0044 | **0.0033** |

## B  Additional motivation behind norm-preserving action

Figure 5 provides an additional example of the difference between the value-preserving and norm-preserving actions. Panel (a) shows two phase functions generated from Prior Model 1. Panel (b) shows the first six modified Fourier basis functions. In (c)&(d), we show the same basis functions after applying the value-preserving and norm-preserving actions using the phase functions in (a). Finally, in (e), we provide an example function formed using a linear combination of the basis functions in (b) (blue), and the same linear combinations but using basis functions in (c) (red) and (d) (yellow), respectively. Note that the norm-preserving action, when combined with an orthonormal basis system, provides more flexibility in modeling. The original blue function as well as the transformed red function contain four extrema in both rows. On the other hand, the yellow functions have five extrema (there is a small local maximum in the yellow function in the bottom row near $t = 0.1$).

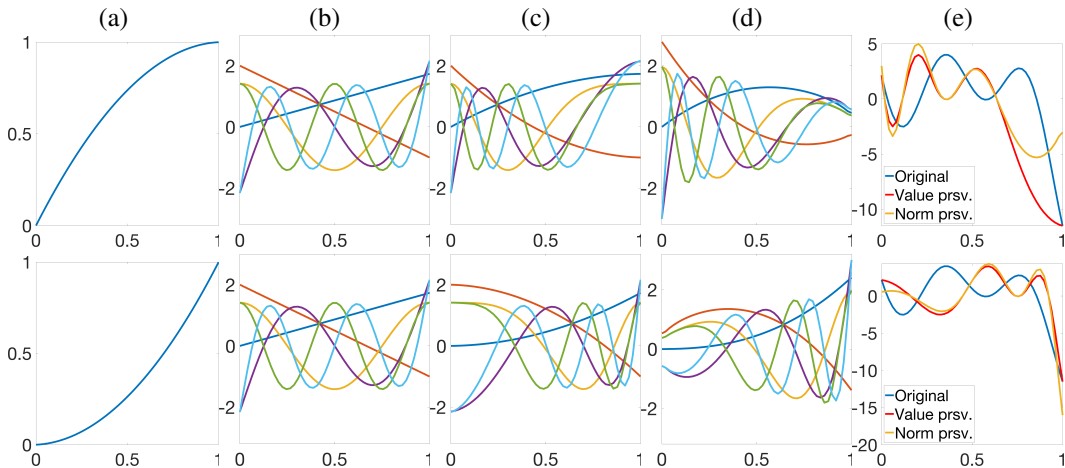

Figure 5: (a) Phase function. (b) Six modified Fourier basis functions. (c) The same basis functions as in (b) after value preserving action using phase function in (a). (d) Same as (c), but using norm-preserving action. (e) Function formed using the same linear combination of basis functions in (b) (blue), (c) (red) and (d) (yellow).

In another example, we study the magnitude of residuals when data is (i) projected onto a fixed number of modified Fourier basis functions, and (ii) projected onto the same number of modified Fourier basis functions followed by optimization over phase functions under the norm-preserving action. We vary the number of basis functions from 1 to 30 and average the residuals over all $n = 93$ Berkeley growth rate functions. Given a function $f \in \mathbb{L}^2$, its projection onto the basis is $\hat{f} = \sum_{j=1}^{B} \hat{a}_j \phi_j$, where $\hat{a}_j = \int_0^1 f(t)\phi_j(t)dt$, $j = 1, \ldots, B$, $B$ is the number of basis functions used in the projection, and $\phi_j$, $j = 1, \ldots, B$ are the basis functions. The residual for (i) is defined as $\|f - \hat{f}\|$, while the residual for (ii) is $\min_{\gamma \in \Gamma} \|f - (\hat{f} \circ \gamma)\sqrt{\dot{\gamma}}\|$. Overall, we expect the average residual from (ii) to be lower than the average residual from (i). However, an interesting aspect of this experiment is to understand how much more expressive the basis is when an additional rotation via the norm-preserving action is allowed, especially when very few basis elements are used in the projection. The results are presented in Figure 6. The average residuals, for $B = 1, \ldots, 30$, based on (i) and (ii) are shown in blue and red. As expected, (ii) yields lower average residuals, with a very large gap for $B = 1, \ldots, 10$. The inclusion of the norm-preserving action becomes less valuable when a large number of basis elements is used. This is also expected since a data-driven rotation of the coordinate system becomes less crucial when more and more coordinates, defined with respect to the basis, are included. This shows that, when a small number of basis elements is used in the specification of $\mu$ in our model, the norm-preserving action allows for more flexible and efficient modeling, by rotating the coordinate system for $\mu$ to the observed data.

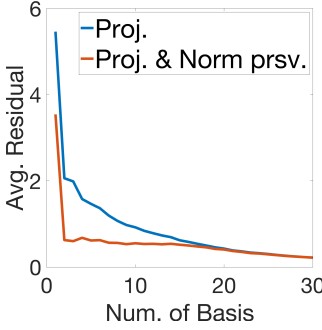

Figure 6: Average residual of (i) projection onto modified Fourier basis functions (blue), and (ii) projection followed by optimization over $\Gamma$ under norm-preserving action (red).

## C  Derivation of marginal likelihood

Conditional on the coefficients of the size-and-shape altering random effect, $\boldsymbol{f}_i$ follows a multivariate normal distribution:

$$\boldsymbol{f}_i|\boldsymbol{c}_i \sim \text{MVN}(\Phi_i\boldsymbol{a} + \tilde{\Phi}_i\boldsymbol{c}_i, \sigma^2\text{diag}(\dot{\gamma}_i(t_j))).$$

The expected value and variance of $\boldsymbol{f}_i$ are

$$\mathbb{E}[\boldsymbol{f}_i] = \mathbb{E}[\mathbb{E}[\Phi_i\boldsymbol{a} + \tilde{\Phi}_i\boldsymbol{c}_i|\boldsymbol{c}_i]] = \mathbb{E}[\Phi_i\boldsymbol{a} + \tilde{\Phi}_i\boldsymbol{c}_i] = \Phi_i\boldsymbol{a} + \tilde{\Phi}_i\mathbb{E}[\boldsymbol{c}_i] = \Phi_i\boldsymbol{a}.$$

and

$$\text{Var}[\boldsymbol{f}_i] = \mathbb{E}[\text{Var}[\boldsymbol{f}_i|\boldsymbol{c}_i]] + \text{Var}[\mathbb{E}[\boldsymbol{f}_i|\boldsymbol{c}_i]] = \sigma^2\text{diag}(\dot{\gamma}_i(t_j)) + \text{Var}[\Phi_i\boldsymbol{a} + \tilde{\Phi}_i\boldsymbol{c}_i] =$$
$$= \sigma^2\text{diag}(\dot{\gamma}_i(t_j)) + \text{Var}[\tilde{\Phi}_i\boldsymbol{c}_i] = \sigma^2\text{diag}(\dot{\gamma}_i(t_j)) + \sigma_c^2\tilde{\Phi}_i\tilde{\Phi}_i^T.$$

Thus, the marginal distribution of $\boldsymbol{f}_i$ is

$$\boldsymbol{f}_i \sim \text{MVN}(\Phi_i\boldsymbol{a}, \sigma^2\text{diag}(\dot{\gamma}_i(t_j)) + \sigma_c^2\tilde{\Phi}_i\tilde{\Phi}_i^T). \tag{9}$$

## D  Proposals and derivation of Metropolis–Hastings acceptance ratio

### D.1  Proposal distributions

We use the following proposal distributions in the Markov chain Monte Carlo algorithm:

1. Fixed effect coefficients: $\boldsymbol{a}^{can} \sim N(\boldsymbol{a}^{cur}, \boldsymbol{\Sigma_a})$.
2. Variance of error process: $(\sigma^2)^{can} \sim \text{TN}((\sigma^2)^{cur}, \tau_\sigma^2, 0, \infty)$.
3. Variance of size-and-shape altering random effect: $(\sigma_c^2)^{can} \sim \text{TN}((\sigma_c^2)^{cur}, \tau_{\sigma_c}^2, 0, \infty)$.
4. Size-and-shape preserving random effect (phase functions) under Prior Model 1: $\alpha_i^{can} \sim$ Uniform$(\alpha_i^{cur} - \delta, \alpha_i^{cur} + \delta)$.
5. Size-and-shape preserving random effect (phase functions) under Prior Model 2: $\gamma_{can} = \gamma_{cur} \circ \tilde{\gamma}$, where $p(\tilde{\gamma}) \sim \text{Dirichlet}(\alpha\boldsymbol{t})$.

TN stands for the truncated normal distribution. Notably, the proposal in 5. for phase functions utilizes the group structure of $\Gamma$. The proposal covariance matrix $\boldsymbol{\Sigma_a}$ is adapted during the burn-in period according to the empirical correlation matrix of the samples of $\boldsymbol{a}$. The proposal variances for the error process and the size-and-shape altering random effect coefficients, $\tau_\sigma^2$ and $\tau_{\sigma_c}^2$, respectively, as well as the parameter $\delta$ and precision parameter $\alpha$ for the size-and-shape preserving random effect under Prior Models 1 and 2, respectively, are adapted according to the acceptance rate during the burn-in period.

### D.2  Derivation of Metropolis–Hastings acceptance ratio

Let $L(\cdot|\cdot)$ denote the likelihood function, and $\pi(\cdot|\cdot)$, $p(\cdot)$ and $q(\cdot|\cdot)$ denote the posterior, prior and proposal distributions. Then, the general form of the MH acceptance ratio for a parameter $\beta_k$, an element of a parameter vector $\boldsymbol{\beta}$, with observed data denoted by $\boldsymbol{f}$, is

$$\rho_{\beta_k} = \min\left\{1, \frac{\pi\left(\beta_k^{can}, \boldsymbol{\beta}_{(-k)}^{cur}|\boldsymbol{f}\right)q\left(\beta_k^{cur}|\beta_k^{can}\right)}{\pi\left(\beta_k^{cur}, \boldsymbol{\beta}_{(-k)}^{cur}, |\boldsymbol{f}\right)q\left(\beta_k^{can}|\beta_k^{cur}\right)}\right\} =$$
$$= \min\left\{1, \frac{L\left(\boldsymbol{f}|\beta_k^{can}, \boldsymbol{\beta}_{(-k)}^{cur}\right)p\left(\beta_k^{can}\right)q\left(\beta_k^{cur}|\beta_k^{can}\right)}{L\left(\boldsymbol{f}|\beta_k^{cur}, \boldsymbol{\beta}_{(-k)}^{cur}\right)p\left(\beta_k^{cur}\right)q\left(\beta_k^{can}|\beta_k^{cur}\right)}\right\}, \tag{10}$$

where $\boldsymbol{\beta}_{(-k)}$ is the vector of parameters excluding $\beta_k$. Note that, for the model proposed in this manuscript $\boldsymbol{\beta} = (\boldsymbol{a}, \sigma_c^2, \sigma^2, \gamma)$.

**Fixed effect coefficients $a$.** Let $\boldsymbol{\beta}_{-a}$ denote the vector of parameters excluding $a$. Since the proposal is a random walk centered at the current value, it is symmetric. Thus,

$$\rho_{\boldsymbol{a}} = \min\left\{ 1, \frac{L(\boldsymbol{f}|a^{can}, \boldsymbol{\beta}_{-a}^{cur})p(a^{can})}{L(\boldsymbol{f}|a^{cur}, \boldsymbol{\beta}_{-a}^{cur})p(a^{cur})} \right\}. \tag{11}$$

**Variance of error process, $\sigma^2$, and variance of shape-and-size altering random effect, $\sigma_c^2$.** Since $\sigma^2$ and $\sigma_c^2$ have the same prior and proposal distributions, we derive the acceptance ratio $\rho_{\sigma^2}$ only, and note that $\rho_{\sigma_c^2}$ is the same. The proposal for these two variance parameters is a truncated normal distribution. Thus,

$$q((\sigma^2)^{cur}|(\sigma^2)^{can}) = \frac{1}{\tau}\frac{\phi\left(\frac{(\sigma^2)^{cur}-(\sigma^2)^{can}}{\tau}\right)}{\Phi\left(\frac{(\sigma^2)^{can}}{\tau}\right)}\mathbb{1}_{\{0<(\sigma^2)^{cur}<\infty\}},$$

$$q((\sigma^2)^{can}|(\sigma^2)^{cur}) = \frac{1}{\tau}\frac{\phi\left(\frac{(\sigma^2)^{can}-(\sigma^2)^{cur}}{\tau}\right)}{\Phi\left(\frac{(\sigma^2)^{cur}}{\tau}\right)}\mathbb{1}_{\{0<(\sigma^2)^{can}<\infty\}},$$

where $\phi(\cdot)$ and $\Phi(\cdot)$ denote the probability density and cumulative distribution functions for the standard normal distribution. As a result,

$$\rho_{\sigma^2} = \min\left\{ 1, \frac{L(\boldsymbol{f}|(\sigma^2)^{can}, \boldsymbol{\beta}_{-\sigma^2}^{cur})p((\sigma^2)^{can})q((\sigma^2)^{cur}|(\sigma^2)^{can})}{L(\boldsymbol{f}|(\sigma^2)^{cur}, \boldsymbol{\beta}_{-\sigma^2}^{cur})p((\sigma^2)^{cur})q((\sigma^2)^{can}|(\sigma^2)^{cur})} \right\} =$$

$$= \min\left\{ 1, \frac{L(\boldsymbol{f}|(\sigma^2)^{can}, \boldsymbol{\beta}_{-\sigma^2}^{cur})p((\sigma^2)^{can})\Phi\left(\frac{(\sigma^2)^{cur}}{\tau}\right)}{L(\boldsymbol{f}|(\sigma^2)^{cur}, \boldsymbol{\beta}_{-\sigma^2}^{cur})p((\sigma^2)^{cur})\Phi\left(\frac{(\sigma^2)^{can}}{\tau}\right)}\mathbb{1}_{\{0<(\sigma^2)^{can}<\infty\}} \right\}. \tag{12}$$

**Size-and-shape preserving random effect $\gamma_i$ under Prior Model 1.** Recall that, under this prior model, $\gamma_i(t) = t + \alpha_i t(1 - t),\ \alpha \in (-1, 1),\ t \in [0, 1]$. The $\text{Unif}(-1, 1)$ prior on $\alpha_i$ assigns zero mass to $\alpha_i^{can} \notin (-1, 1)$ resulting in automatic rejection. In other cases, the proposal $\text{Unif}(\alpha_i^{cur} - \delta, \alpha_i^{cur} + \delta)$ is symmetric. Thus,

$$\rho_{\alpha_i} = \min\left\{ 1, \frac{L(\boldsymbol{f}|\alpha_i^{can}, \boldsymbol{\beta}_{-\alpha_i}^{cur})}{L(\boldsymbol{f}|\alpha_i^{cur}, \boldsymbol{\beta}_{-\alpha_i}^{cur})}\mathbb{1}_{\{-1<\alpha_i^{can}<1\}} \right\}. \tag{13}$$

**Size-and-shape preserving random effect $\gamma_i$ under Prior Model 2.** Since

$$\rho_{\gamma_i} = \min\left\{ 1, \frac{L(\boldsymbol{f}|\gamma_i^{can}, \boldsymbol{\beta}_{-\gamma_i}^{cur})p(\gamma_i^{can})q(\gamma_i^{cur}|\gamma_i^{can})}{L(\boldsymbol{f}|(\gamma_i^{cur}, \boldsymbol{\beta}_{-\gamma_i}^{cur})p(\gamma_i^{cur})q(\gamma_i^{can}|\gamma_i^{cur})} \right\}, \tag{14}$$

we focus on the derivation of $\frac{q(\gamma_i^{cur}|\gamma_i^{can})}{q(\gamma_i^{can}|\gamma_i^{cur})}$. We omit the subscript $i$ to simplify notation. Let

$$p(\gamma) = (\gamma(t_2) - \gamma(0), \gamma(t_3) - \gamma(t_2), \ldots \gamma(t_{T_\gamma-1}) - \gamma(t_{T_\gamma-2}), \gamma(1) - \gamma(t_{T_\gamma-1})) =$$
$$=: (\Delta_1(\gamma), ..., \Delta_{T_\gamma-1}(\gamma)) =: \Delta(\gamma) \tag{15}$$

The proposal distribution is $\Delta(\tilde{\gamma}) \sim \text{Dirichlet}(\alpha\boldsymbol{t})$, where $\boldsymbol{t} = (t_2, t_3 - t_2, \ldots, 1 - t_{T_\gamma-1}) = (t_{(1)}, \ldots, t_{(T_\gamma-1)})$, and has density

$$\frac{\Gamma(\alpha)}{\prod_{j=1}^{T_\gamma-1}\Gamma(\alpha t_{(j)})}\prod_{j=1}^{T_\gamma-1}(\Delta_j(\tilde{\gamma}))^{\alpha t_{(j)}-1}. \tag{16}$$

Next, we derive the density of $\Delta(\gamma_{can})$ given $\Delta(\tilde{\gamma})$ and $\gamma_{cur}$, which appears in the denominator of the acceptance ratio. Since $\tilde{\gamma} = \gamma_{cur}^{-1} \circ \gamma_{can}$, we have

$$\Delta(\tilde{\gamma}) = \Big( \tilde{\gamma}(t_2), \ldots, \tilde{\gamma}(t_3) - \tilde{\gamma}(t_2), \ldots, \tilde{\gamma}(1) - \tilde{\gamma}(t_{T_\gamma - 1}) \Big) =$$

$$= \Big( \gamma_{cur}^{-1}(\gamma_{can}(t_2)), \gamma_{cur}^{-1}(\gamma_{can}(t_3)) - \gamma_{cur}^{-1}(\gamma_{can}(t_2)), \ldots, \gamma_{cur}^{-1}(\gamma_{can}(1)) - \gamma_{cur}^{-1}(\gamma_{can}(t_{T_\gamma - 1})) \Big) =$$

$$= \Big( \gamma_{cur}^{-1}(\Delta_1(\gamma_{can})), \gamma_{cur}^{-1}\Big( \sum_{j=1}^{2} \Delta_j(\gamma_{can}) \Big) - \gamma_{cur}^{-1}(\Delta_1(\gamma_{can})),$$

$$\gamma_{cur}^{-1}\Big( \sum_{j=1}^{3} \Delta_j(\gamma_{can}) \Big) - \gamma_{cur}^{-1}\Big( \sum_{j=1}^{2} \Delta_j(\gamma_{can}) \Big) \ldots,$$

$$\gamma_{cur}^{-1}\Big( \sum_{j=1}^{T_\gamma - 1} \Delta_j(\gamma_{can}) \Big) - \gamma_{cur}^{-1}\Big( \sum_{j=1}^{T_\gamma - 2} \Delta_j(\gamma_{can}) \Big) \Big) =$$

$$=: g(\Delta(\gamma_{can}); \gamma_{cur}). \tag{17}$$

Therefore, the density of $\Delta(\gamma_{can})$ given $\Delta(\tilde{\gamma})$ and $\gamma_{cur}$ is the density of $\Delta(\tilde{\gamma})$, i.e., the density of Dirichlet$(\alpha t)$, multiplied by the determinant of the Jacobian of $g$. The Jacobian of $g$ is given by

$$\frac{\partial g}{\partial(\Delta(\gamma_{can}))} = \begin{bmatrix} \frac{\partial g_1}{\partial \Delta_1} & \cdots & \frac{\partial g_1}{\partial \Delta_{T_\gamma - 1}} \\ \frac{\partial g_2}{\partial \Delta_1} & \cdots & \frac{\partial g_2}{\partial \Delta_{T_\gamma - 1}} \\ \vdots & \ddots & \vdots \\ \frac{\partial g_{T_\gamma - 1}}{\partial \Delta_1} & \cdots & \frac{\partial g_{T_\gamma - 1}}{\partial \Delta_{T_\gamma - 1}} \end{bmatrix} =$$

$$= \begin{bmatrix} \dot{\gamma}_{cur}^{-1}(\Delta_1(\gamma_{can})) & 0 & \cdots & 0 \\ \cdots & \dot{\gamma}_{cur}^{-1}(\sum_{j=1}^{2} \Delta_j(\gamma_{can})) & \cdots & 0 \\ \vdots & \vdots & \ddots & \vdots \\ \cdots & \cdots & \cdots & \dot{\gamma}_{cur}^{-1}(\sum_{j=1}^{T_\gamma - 1} \Delta_j(\gamma_{can})) \end{bmatrix}, \tag{18}$$

and is a lower triangular matrix. Then, the determinant of this Jacobian is

$$\left| \frac{\partial g}{\partial(\Delta(\gamma_{can}))} \right| = \prod_{j=1}^{T_\gamma - 1} \dot{\gamma}_{cur}^{-1}\Big( \sum_{k=1}^{j} \Delta_k(\gamma_{can}) \Big). \tag{19}$$

The numerator of the acceptance ratio contains the density of $\Delta(\gamma_{cur})$, given $\Delta(\tilde{\gamma}^{-1})$ and $\gamma_{can}$, which is the density of $\Delta(\tilde{\gamma}^{-1})$ multiplied by the determinant of the Jacobian of $h := h(\Delta(\gamma_{cur}); \tilde{\gamma}^{-1})$, the transformation from $\Delta(\gamma_{cur})$ to $\Delta(\tilde{\gamma}^{-1})$. This determinant is derived in the same way as the determinant of the Jacobian of $g$. Finally, we have

$$\frac{q(\gamma^{cur}|\gamma^{can})}{q(\gamma^{can}|\gamma^{cur})} = \frac{\frac{\Gamma(\alpha)}{\prod_{j=1}^{T_\gamma - 1} \Gamma(\alpha t_{(j)})} \prod_{j=1}^{T_\gamma - 1}(\Delta_j(\tilde{\gamma}^{-1}))^{\alpha t_{(j)} - 1} \prod_{j=1}^{T_\gamma - 1} \dot{\gamma}_{can}^{-1}\Big( \sum_{k=1}^{j} \Delta_k(\gamma_{cur}) \Big)}{\frac{\Gamma(\alpha)}{\prod_{j=1}^{T_\gamma - 1} \Gamma(\alpha t_{(j)})} \prod_{j=1}^{T_\gamma - 1}(\Delta_j(\tilde{\gamma}))^{\alpha t_{(j)} - 1} \prod_{j=1}^{T_\gamma - 1} \dot{\gamma}_{cur}^{-1}\Big( \sum_{k=1}^{j} \Delta_k(\gamma_{can}) \Big)}. \tag{20}$$

## E  Detailed Markov chain Monte Carlo algorithm

The detailed MCMC algorithm is given in Algorithm 1.

---

**Algorithm 1** Bayesian Functional Mixed Effects Model

---

**Input:** Data: $\boldsymbol{f}_i = f_i(t_j)$, $i = 1, \ldots, n$, $j = 1, \ldots, T$; prior hyperparameters: $\tau_a^2$ (prior variance for $\boldsymbol{a}$), $a_\sigma$, $b_\sigma$ (shape and scale for $\sigma^2$), $a_{\sigma_c}$, $b_{\sigma_c}$ (shape and scale for $\sigma_c^2$), $\theta_\gamma$, $\boldsymbol{t}$ (concentration and discretization in Prior Model 2 for $\gamma$); proposal hyperparameters: $\boldsymbol{\Sigma_a}$ (proposal covariance for $\boldsymbol{a}$), $\tau_\sigma^2$ (proposal variance for $\sigma^2$), $\tau_{\sigma_c}^2$ (proposal variance for $\sigma_c^2$), $\delta$ (proposal for $\alpha$ in Prior Model 1 for $\gamma$), $\alpha, \boldsymbol{t}$ (concentration and discretization in Prior Model 2 proposal for $\gamma$); number of

burn-in iteration: $N_b$; total number of MCMC iterations: $N$; number of iterations between tuning of proposal parameters: $N_t$; initial values: $\boldsymbol{a}_0$, $(\sigma^2)_0$, $(\sigma_c^2)_0$, $(\alpha_i)_0$, $i = 1, \ldots, n$ (Prior Model 1) or $(\gamma_i)_0$, $i = 1, \ldots, n$ (Prior Model 2).

**Output:** Posterior samples of all parameters: $\boldsymbol{a}_k$, $(\sigma^2)_k$, $(\sigma_c^2)_k$, $(\alpha_i)_k$, $i = 1, \ldots, n$ (Prior Model 1) or $(\gamma_i)_k$, $i = 1, \ldots, n$ (Prior Model 2), $k = 1, \ldots, N - N_b$.

**for** $j$ in 1:$N$ **do**

    **if** $\mathrm{mod}(j, N_t) == 0$ and $j < N_b$ **then**

        1. Update $\boldsymbol{\Sigma_a}$ based on the empirical correlation matrix of the last $N_t$ samples of $\boldsymbol{a}$.

    **end if**

    2. Propose $\boldsymbol{a}^{can}$ and compute $\rho_{\boldsymbol{a}}$ using (11).

    **if** $U < \rho_{\boldsymbol{a}}$, $U \sim \mathrm{Unif}(0, 1)$ **then**

        3. Set $\boldsymbol{a}_{j+1} = \boldsymbol{a}^{can}$.

    **else**

        3. Set $\boldsymbol{a}_{j+1} = \boldsymbol{a}_j$.

    **end if**

    **if** $\mathrm{mod}(j, N_t) == 0$ and $j < N_b$ **then**

        4. Update $\tau_\sigma^2$ based on the acceptance rate of the last $N_t$ samples of $\sigma^2$.

    **end if**

    5. Propose $(\sigma^2)^{can}$ and compute $\rho_{\sigma^2}$ using (12).

    **if** $U < p_{\sigma^2}$, $U \sim \mathrm{Unif}(0, 1)$ **then**

        6. Set $(\sigma^2)_{j+1} = (\sigma^2)^{can}$.

    **else**

        6. Set $(\sigma^2)_{j+1} = (\sigma^2)_j$.

    **end if**

    **if** $\mathrm{mod}(j, N_t) == 0$ and $j < N_b$ **then**

        7. Update $\tau_{\sigma_c}^2$ based on the acceptance rate of the last $N_t$ samples of $\sigma_c^2$.

    **end if**

    8. Propose $(\sigma_c^2)^{can}$ and compute $\rho_{\sigma_c^2}$ using (12).

    **if** $U < p_{\sigma_c^2}$, $U \sim \mathrm{Unif}(0, 1)$ **then**

        9. Set $(\sigma_c^2)_{j+1} = (\sigma_c^2)^{can}$.

    **else**

        9. Set $(\sigma_c^2)_{j+1} = (\sigma_c^2)_j$.

    **end if**

    **for** $i$ in 1:$n$ **do**

        **if** Prior Model 1 **then**

            **if** $\mathrm{mod}(j, N_t) == 0$ and $j < N_b$ **then**

                10. Update $\delta$ based on the acceptance rate of the last $N_t$ samples of $\alpha_i$.

            **end if**

            11. Propose $\alpha_i^{can}$ and compute $\rho_{\alpha_i}$ using (13).

            **if** $U < p_{\alpha_i}$, $U \sim \mathrm{Unif}(0, 1)$ **then**

                12. Set $(\alpha_i)_{j+1} = \alpha_i^{can}$.

            **else**

                12. Set $(\alpha_i)_{j+1} = (\alpha_i)_j$.

            **end if**

        **end if**

        **if** Prior Model 2 **then**

            **if** $\mathrm{mod}(j, N_t) == 0$ and $j < N_b$ **then**

                10. Update $\alpha$ based on the acceptance rate of the last $N_t$ samples of $\gamma_i$.

            **end if**

            11. Propose $\gamma_i^{can}$ and compute $\rho_{\gamma_i}$ using (14).

            **if** $U < p_{\gamma_i}$, $U \sim \mathrm{Unif}(0, 1)$ **then**

                12. Set $(\gamma_i)_{j+1} = \gamma_i^{can}$.

            **else**

                12. Set $(\gamma_i)_{j+1} = (\gamma_i)_j$.

            **end if**

        **end if**

    **end for**

**end for**

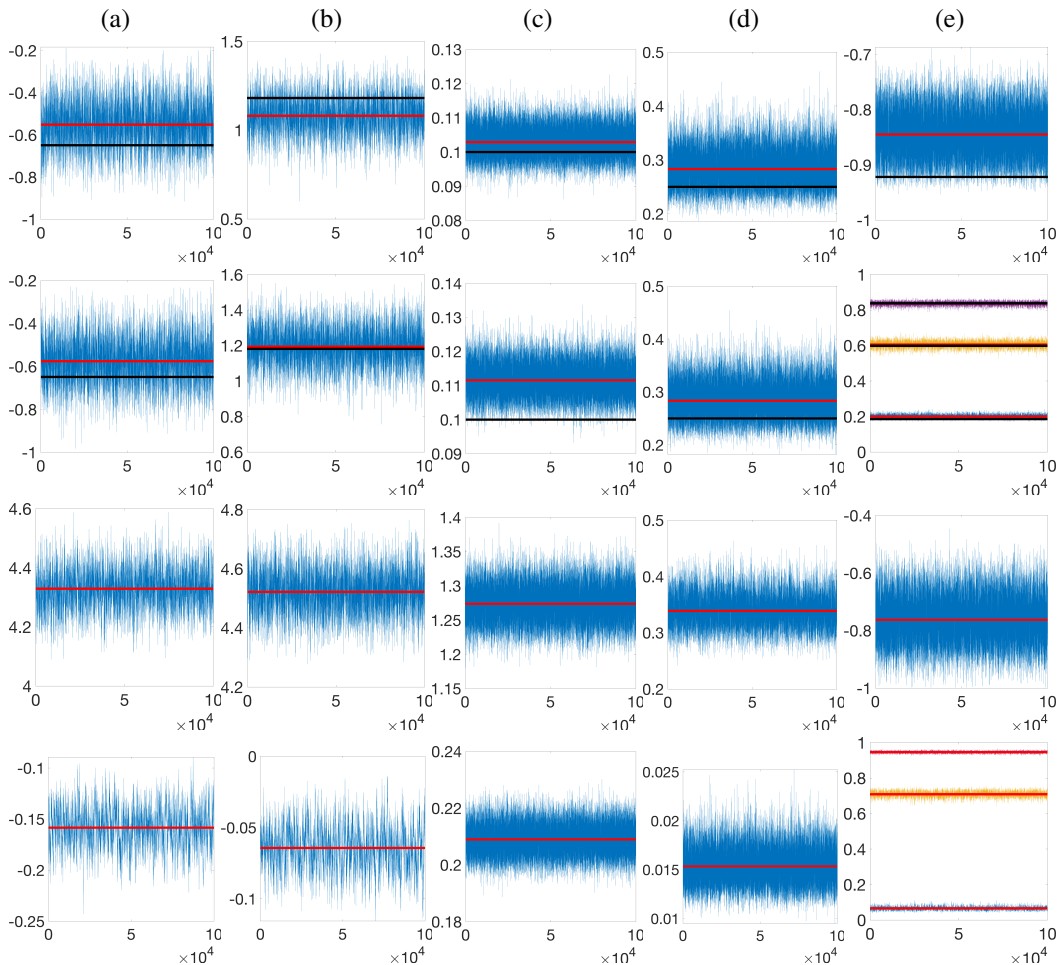

Figure 7: Row 1: Simulated data - row 1 in Figure 2 in Section 4.1. Row 2: Simulated data - row 2 in Figure 2 in Section 4.1. Row 3: Berkeley - row 1 in Figure 4 in Section 4.2. Row 4: PQRST - row 2 in Figure 4 in Section 4.2. Trace plots for the (a)&(b) first two fixed effect coefficients in $\boldsymbol{a}$, respectively, (c) error process variance $\sigma^2$, (d) variance of size-and-shape altering random effect $\sigma_c^2$, and (e) size-and-shape preserving random effect (phase function) for a randomly chosen observation. Ground truth and posterior mean are marked in black and red, respectively.

# F MCMC diagnostic plots for examples in Section 4

Figure 7 shows trace plots of $100,000$ MCMC iterations after the burn-in period for all model parameters. The first two rows correspond to the simulated data examples considered in Figure 2 in Section 4.1. The third and fourth rows correspond to the two real data examples considered in Figure 4 Section in 4.2 (third row: Berkeley; fourth row: PQRST). Panels (a) and (b) show trace plots for the first two coefficients (first two entries in the vector $\boldsymbol{a}$ of the fixed effect function $\mu$. Panels (c) and (d) show trace plots for $\sigma^2$ and $\sigma_c^2$, respectively. Finally, panel (e) shows trace plots for the parameter $\alpha_i$ when Prior Model 1 was used, or the parameter values $\gamma_i(t_2)$ (blue), $\gamma_i(t_3)$ (yellow) and $\gamma_i(t_4)$ (purple) when Prior Model 2 was used (recall that $t_2 = 0.25$, $t_3 = 0.5$ and $t_4 = 0.75$). The choices of phase functions $\gamma_i$ in panel (e) match those in Sections 4.1 and 4.2. For the simulated examples, we show the ground truth value of each parameter as a horizontal black line. For all examples, we show the posterior mean of each parameter, estimated using the $100,000$ samples shown in the trace plots, using a horizontal red line. All trace plots suggest convergence to the stationary posterior distribution.

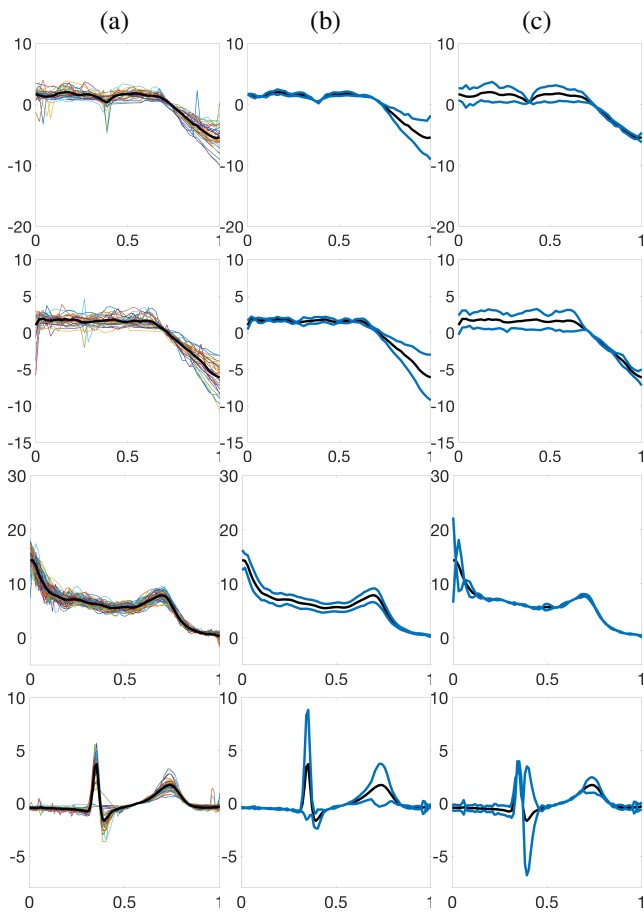

Figure 8: Top to bottom: data simulated using Prior Model 1 for phase functions, data simulated using Prior Model 2 for phase functions, Berkeley growth rate functions and PQRST complexes. (a) Functions $(f_i \circ \gamma_i^*)\sqrt{\dot{\gamma}_i^*}$ and average $\bar{\mu}$ (black). (b) Average $\bar{\mu}$ (black) and $\bar{\mu} \pm U_1$ (blue). (c) Same as (b), but using $U_2$.

## G  Empirical FPCA basis for fixed effect function

Here, we provide empirical evidence behind the claim that the use of an FPCA basis deteriorates estimation performance for $\mu$ (see Contribution 3 in Section 1). We demonstrate this using four datasets considered in Section 4: the two simulated datasets from Example 1 in Section 4.1 and the two real datasets from Section 4.2. To estimate the FPCA basis, we follow the following steps: (i) estimate the (centered) sample average defined as $\bar{\mu} = \arg\min_{\mu \in \mathbb{L}^2} \sum_{i=1}^{n} \min_{\gamma \in \Gamma} \|\mu - (f_i \circ \gamma_i)\sqrt{\dot{\gamma}_i}\|^2$, (ii) compute the sample covariance $K$ of $w_i = (f_i \circ \gamma_i^*)\sqrt{\dot{\gamma}_i^*} - \bar{\mu}$ where $\gamma_i^* = \arg\min_{\gamma \in \Gamma} \|\mu - (f_i \circ \gamma_i)\sqrt{\dot{\gamma}_i}\|^2$ (assuming that each $w_i$ is sampled at $t_1 = 0,\ t_2, \ldots,\ t_{T-1},\ t_T$), and (iii) apply singular value decomposition to the covariance matrix $K = USU^\top$. The columns $U$ provide a data-driven orthonormal FPCA basis for $\mathbb{L}^2$.

The number of FPCA basis functions $B_f$ used to specify $\mu$ in the proposed Bayesian model is selected based on $\%$ variation explained: $90\%$ for the two simulated datasets ($B_f = 7$ and $B_f = 5$ for data simulated under Prior Models 1 and 2 for phase functions, respectively) and for the PQRST complexes ($B_f = 6$); $80\%$ for the Berkeley growth rate functions $B_f = 8$. In all cases, we use $B_r = 6$ B-spline basis functions for the shape-and-size altering random effect. We use the same prior models for phase functions for each dataset as those that were used in Section 4.

First, in Figure 8, we show results of applying the FPCA procedure to each dataset. Panel (a) shows the functions $(f_i \circ \gamma_i^*)\sqrt{\dot{\gamma}_i^*}$, and $\bar{\mu}$ in black. Panels (b)&(c) display the variation in the data captured by the two leading FPCA basis functions: $\bar{\mu}$ in black with $\bar{\mu} \pm U_j,\ j = 1, 2$ in blue ($j = 1$ in (b)

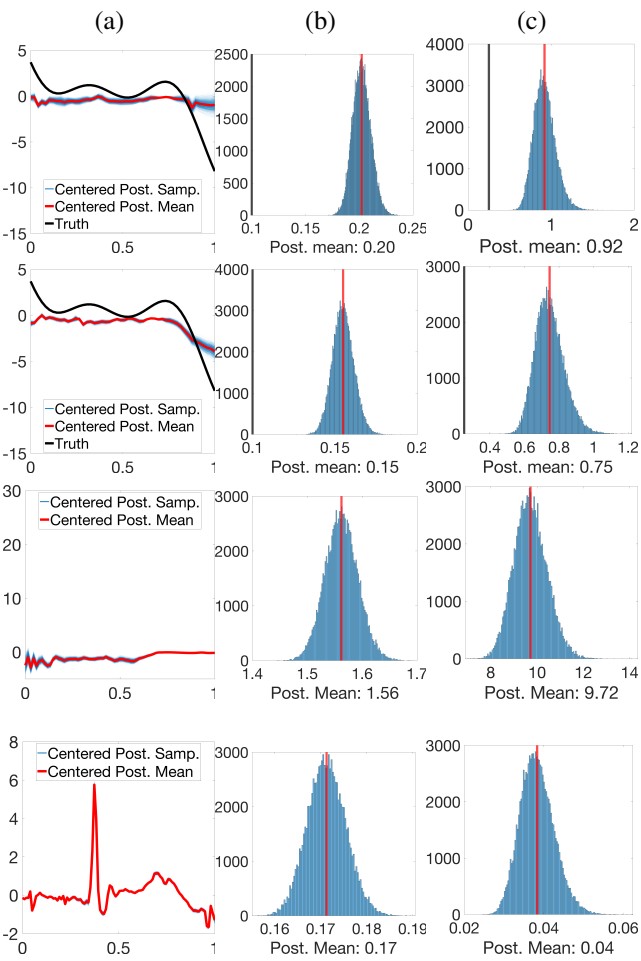

Figure 9: Top to bottom: data simulated using Prior Model 1 for phase functions, data simulated using Prior Model 2 for phase functions, Berkeley growth rate functions and PQRST complexes. Rows 1&3: model with Prior Model 1 on $\Gamma$. Rows 2&4: model with Prior Model 2 on $\Gamma$. (a) Estimation of $\mu$: ground truth when available (black), centered posterior samples (blue), centered posterior mean (red). (b)&(c) Histograms of posterior samples for $\sigma^2$ and $\sigma_c^2$, respectively (posterior mean in red; ground truth in black).

and $j = 2$ in (c)). Notably, the FPCA basis appears to capture various sources of variation including noise, which is undesirable if they are to be used to model the fixed effect function $\mu$.

Figure 9 provides estimation results for (a) $\mu$ (ground truth in black when available, centered posterior samples in blue, and centered posterior mean in red), (b) $\sigma^2$ (ground truth in black when available, posterior mean in red), and (c) $\sigma_c^2$ (ground truth in black when available, posterior mean in red). It is clear that the data-driven FPCA basis does not provide a good model for the fixed effect function $\mu$. Only for the PQRST data example, the model yields a reasonable estimate of $\mu$. This in turn results in overestimation of $\sigma^2$ and $\sigma_c^2$ (posterior samples of $\sigma_c^2$ are very large for the Berkeley data). This suggests that most variation is absorbed into the size-and-shape altering random effect and observation error.

## H  Sensitivity analysis for hyperparameter misspecification

We assess sensitivity of posterior inference to under- or over-specification of three hyperparameters in the proposed model: $B_f$ (number of basis functions used to model the fixed effect function $\mu$), $B_r$ (number of basis functions used to model the size-and-shape altering random effect $v_i$) and $\theta_\gamma$ (concentration hyperparameter in Prior Model 2 (PM 2) on the size-and-shape preserving random

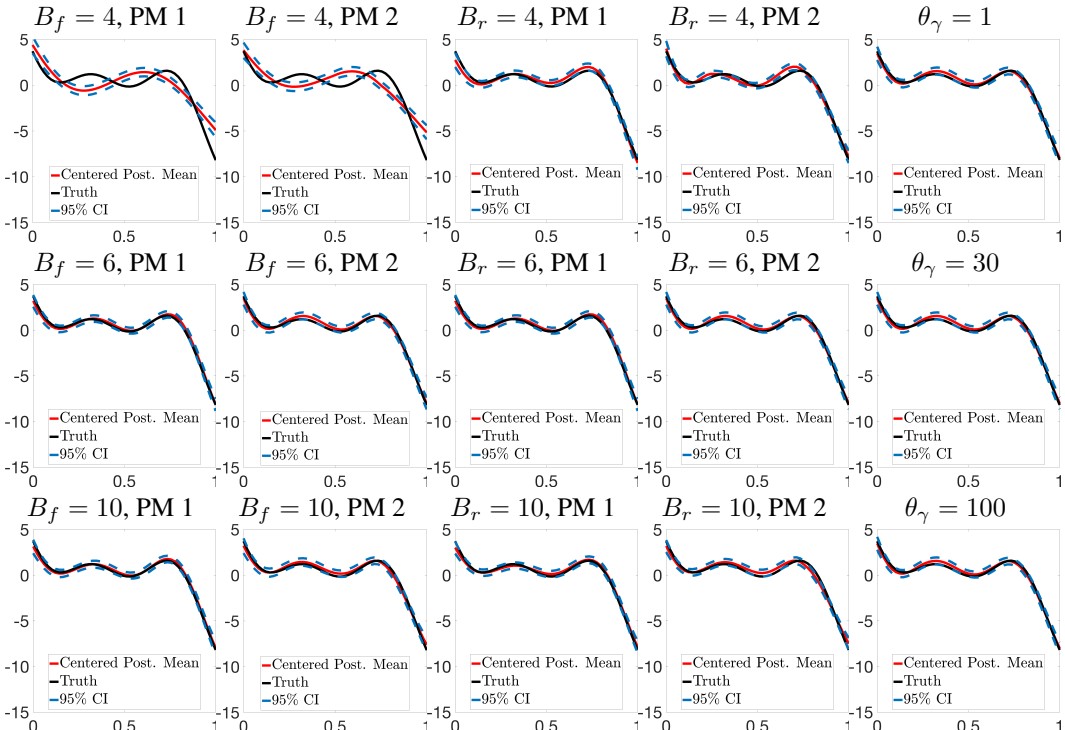

Figure 10: Centered posterior mean (red) and 95% credible interval (dashed blue) for $\mu$ (ground truth, black) for different choices of $B_f$, $B_r$ and $\theta_\gamma$ in Prior Model 2 (PM 2) on phase functions; PM 1 refers to Prior Model 1 on phase. The data was generated using $B_f = B_r = 6$ and $\theta_\gamma = 30$ (for PM 2). Row 1: Estimation results for under-specified values of hyperparameters. Row 2: Estimation results for correctly specified values of hyperparameters. Row 3: Estimation results for over-specified values of hyperparameters.

effect or phase function $\gamma_i$). The data for this experiment is exactly the same as in Example 1 in Section 4.1, i.e., the data was generated from the proposed model with $B_f = 6$ modified Fourier basis functions for $\mu$ and $B_r = 6$ B-spline basis functions for each $v_i$. Each panel in Figure 10 shows the centered estimate of the posterior mean for $\mu$ (red), associated 95% credible interval (dashed blue), and the ground truth $\mu$ (black). Rows 1-3 show results for under-specified, correctly specified and over-specified values of the three hyperparameters, respectively. Overall, posterior inference, in terms of the posterior mean for $\mu$ and its uncertainty as ascertained via the 95% credible interval, is very robust to (i) over-specification of $B_f$, (ii) under- or over-specification of $B_r$, and (iii) under- or over-specification of $\theta_\gamma$. On the other hand, when $B_f$ is under-specified, we are unable to accurately recover $\mu$ as seen in row 1. This is not unexpected since the ground truth $\mu$ does not lie in the subspace spanned by the specified basis functions. Thus, in general, we recommend specifying a larger number of basis functions to model the fixed and size-and-shape altering random effects.

# I   Additional real data examples

We provide estimation results for $\mu$ for several additional datasets that have been analyzed in previous literature:

1. Pinch force data [Ramsay et al., 1995], which was also analyzed by Claeskens et al. [2021].

2. Respiration data [Kurtek et al., 2013].

3. Gait data [Kurtek et al., 2013].

4. Signature acceleration data [Kneip and Ramsay, 2008]

5. Gene expression data [Srivastava et al., 2011b].

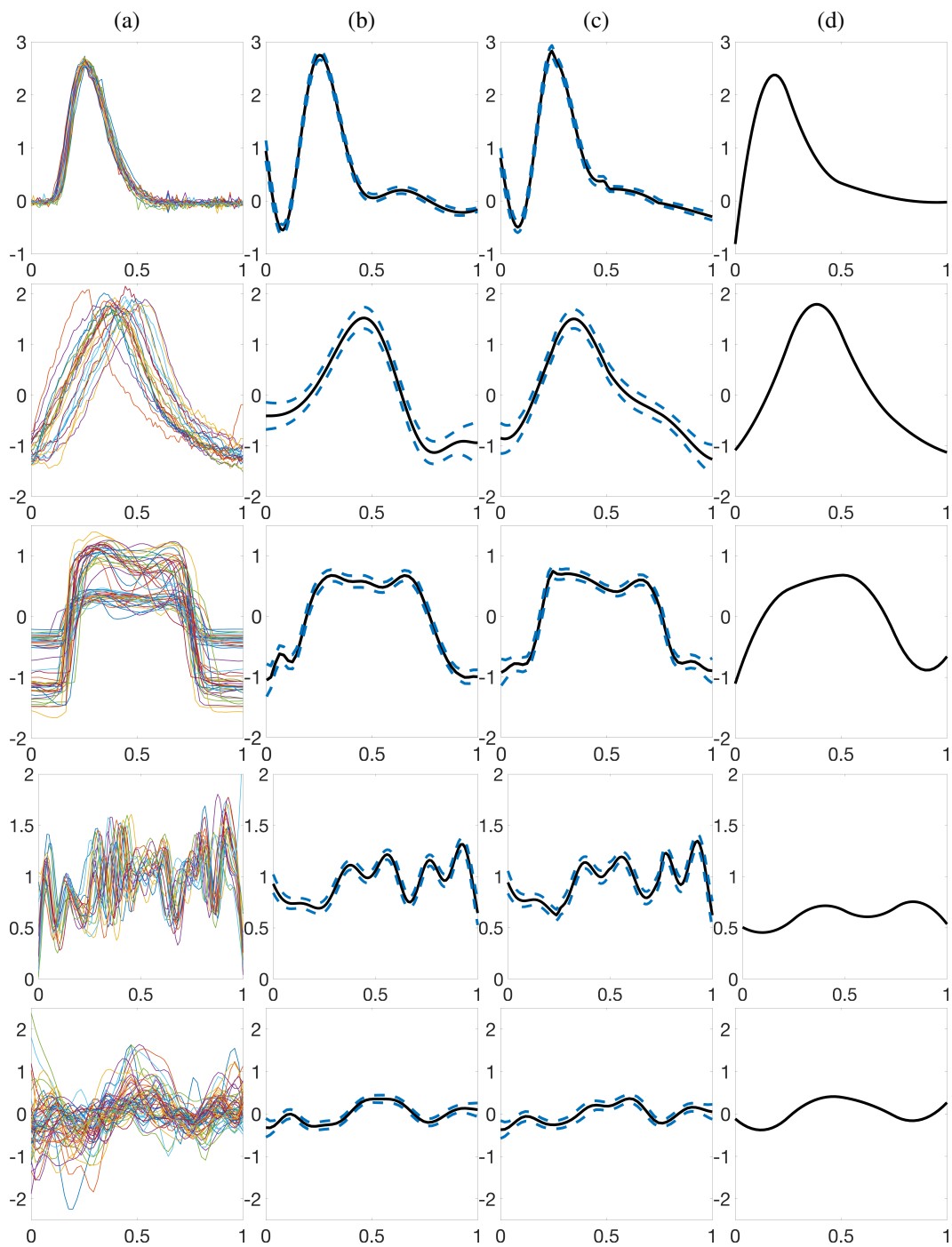

Figure 11: Estimation results for the pinch force, respiration, gait, signature acceleration and gene expression datasets (top to bottom). (a) Data. (b)&(c) Centered posterior mean (black) and 95% credible interval (dashed blue) for $\mu$ when Prior Models 1 and 2 are used for phase functions, respectively. (d) `warpMix` estimate.

We omit detailed descriptions of these datasets here and refer the interested reader to the associated references.

To specify the model for the fixed effect function $\mu$, we use the modified Fourier basis with $B_f = 6$ basis functions for the pinch force and respiration datasets, and $B_f = 12$ basis functions for the gait,

signature acceleration and gene expression datasets. We use $B_r = 6$ B-spline basis functions for each $v_i$ in all cases.

Figure 11(a) displays each dataset (top to bottom: pinch force, respiration, gait, signature acceleration, gene expression). Panels (b) and (c) in Figure 11 show estimation results for $\mu$ when Prior Models 1 and 2 are used for phase functions, respectively. We show the centered posterior mean in black with a 95% credible interval displayed using blue dashed lines. The credible interval is computed pointwise, using the 2.5% and 97.5% empirical quantiles of the centered posterior samples, $(\hat{\mu}^j \circ \bar{\gamma})\sqrt{\dot{\bar{\gamma}}}$, $j = 1, \ldots, N$. The warpMix estimate is shown in Figure 11(d). Overall, the proposed Bayesian model is effective in recovering $\mu$. Compared to warpMix, our estimates contain finer geometric features and are more representative of the size-and-shape patterns in the data.

