# OpenReview forum: "Probabilistic size-and-shape functional mixed models"
_NeurIPS.cc/2024/Conference — NeurIPS 2024 poster_

### Official Review · Reviewer_UDtq · 2024-07-02

**Soundness:** 3
**Presentation:** 3
**Contribution:** 2
**Rating:** 7
**Confidence:** 4

**Summary:**

The paper deals with estimation functional data, i.e. time-series represented as functions – under a specific observation model, that aims to separately account for a fixed effect (modeled as a mean function) and other variations (modeled as noise functions), with the added confounding variable being a norm-preserving warping function, that leads to observed data.

The paper proceeds to develop an Bayesian approach to estimating the model parameters, under certain assumed distributions for the noise terms, and certain basis functions to represent the functions in a finite dimensional space.

Evaluation is shown both on simulated data and some real functional data examples.

**Strengths:**

The paper does a good job of introducing various temporal warping models, and their intuition on many example data. I found the mathematical and visual explanations of the differences between value preserving and norm preserving warps illuminating. The paper is well positioned to advance the field of functional data analysis as more and more applications get impacted by these methods.

The paper tries to motivate the presented mixed effects models by highlighting historical examples based on 2D shape analysis, which provides good context for the proposed developments.

The paper also has good strategies for actually implementing the infinite dimensional computations using projections onto basis functions such as B-splines.

**Weaknesses:**

Comparisons are limited to a publicly available implementation of the technique known as warpMiX. Default parameters are used as given in the R implementation. At the very least some attempt at matching noise variance assumed in the warpmix model and the proposed model should be attempted. Default noise variance parameter in warpmix is at the level of 10^-3 whereas the noise variances for the proposed model are much higher at sigma_c^2 = 0.25, sigma^2 = 1 etc… Are these comparable in any sense, if not how should this comparison be made more fairly?

Further, the paper cites a wealth of literature that uses a model similar to the one used, at least as far as the assumed norm preserving action model. There are papers by the cited authors that use these models for estimating mean functions from a stack of observed functions, mainly by Kurtek, Srivastava and colleagues. Many of these references are also cited in the paper, but some attempt should be made to compare to these approaches.

Beyond simulation, for the task of recovery of functions (and functional parameters) under an assumed observation model, evaluation should include some real downstream task. Otherwise, we are only faced with visual assessments of quality. For instance, the results as shown in figure 4 are claimed to be better in the sense that the estimate is ‘as desired and contains sharp features that are representative of the observed data’ (line 351, pg 9). These are at best subjective assessments, and it is unclear if such an estimate is useful in some quantifiable way. There are many ways to take this to the next step of evaluation, to see if some downstream task such as classification shows improvement after the recovery of the needed parameters.

**Questions:**

I would like to know how well the method could be implemented without using basis function approximations – i.e. representing functions simply by their samples. What gets simplified and what gets complicated. What are the practical implications of assuming a specific family of basis functions such as B-Splines. I did notice the data-driven functional PCA basis was tried as well, in the appendix, which seemed to perform worse than B-splines.

Additional questions are on evaluation, which are described in the experiments section of the review.

**Limitations:**

The paper has a reasonable description of the limitations of the model and its assumptions in section 3.

---

> ### Author Rebuttal · Authors · 2024-08-05
>
> **Noise Variance** The warpMix default noise variance parameter, "sigmaEpsilonTilde", which is set to $10^{-3}$ in the R implementation, is the variance of $\theta_i$, a set of parameters in the model for the phase functions (Equation (6) in Claeskens et al., Nonlinear Mixed Effects Modeling and Warping for Functional Data, 2021). It is not the noise variance or the variance of the random effect ($\sigma^2$ and $\sigma^2_c$ in our model). In Simulated Example 2 in Sec. 4.1, we compare the proposed Bayesian model to warpMix. There, we actually simulate data using the warpMix model (with default parameter values). Note that, as in the warpMix model, the data is generated using the value-preserving action and not the norm-preserving action used in our model. Despite this, our model recovers the underlying fixed effect function $\mu$ with higher accuracy than the warpMix model (see Table 1 in the manuscript and attached file). In this regard, all of our comparisons to warpMix are fair.
>
> **Novelty and Comparison to Existing Methods** To the best of our knowledge, the Bayesian functional mixed effects model proposed in our manuscript is the first to use the norm-preserving action as a size-and-shape preserving random effect on the original function space. The cited papers use the norm-preserving action on a transformed space, the space of square-root velocity functions, which corresponds to the value-preserving action on the original space (see Chapter 4 in Srivastava et al., Functional and Shape Data Analysis, 2016). In addition, none of those models incorporate size-and-shape altering random effects. The table in the attached file includes a quantitative comparison of the proposed modeling framework to the model of Cheng et al., Bayesian Registration of Functions and Curves, 2016 which is a mean+noise model and utilizes the aforementioned square-root velocity representation. As can be seen in this table, our model outperforms their approach, labeled BRFC, in terms of recovery of the mean (fixed effect) function $\mu$ on all simulated data examples.
>
> **Evaluation of Estimation Results and Novelty** The question of whether the fixed effect function $\mu$ can be reliably recovered is a highly studied problem within the functional data analysis literature, and sufficient conditions have only recently been identified. In this context, our conclusion that the geometric size-and-shape of $\mu$ is recoverable is novel. This finding has clear implications in many machine learning applications with functional data as input to neural networks, which additionally incorporate a random effect in its architecture (Simchoni et al., Integrating Random Effects in Deep Neural Networks, 2023). However, incorporating downstream tasks, e.g., a classification layer, into the proposed model is something of interest and we plan to pursue this in future work. Our quantitative evaluations are based on simulated datasets where the true underlying fixed effect function $\mu$ is known. In these cases, we outperform state-of-the-art competitors including warpMix and BRFC as seen in the table in the attached file. As mentioned by the reviewer, we rely on qualitative evaluations for the real data examples. Nonetheless, in most real data scenarios it is fairly clear that the fixed effect function estimated using the proposed model is "better" or more representative of patterns in the observed data than that recovered using warpMix; see Sec. 4.2 and App. H. For the PQRST complexes, the warpMix model fails to yield an estimate, and for many of the other datasets it tends to oversmooth expected prominent features of $\mu$.
>
> **Alternative Model Formulation** The fixed effect function and size-and-shape altering random effect components of our probability model are specified using linear combinations of basis functions, *but not the discretized data itself*. The main motivation behind this choice is dimension reduction (sample size $n$ is typically smaller than number of time points per function), and this is the most common approach in specifying models for functional data. That said, an alternative approach would be to specify the model pointwise using parameter function evaluations as suggested by the reviewer. This, however, would drastically increase the dimension of the parameter space for the fixed effect function, complicating inference. The basis functions are only used to specify the prior distributions for the fixed effect function and the size-and-shape altering random effect. In that sense, basis functions allow us to enforce desired smoothness in the estimate of $\mu$, which would be much more difficult if the fixed effect function was modeled pointwise. As mentioned by the reviewer, we tried to use a data-driven FPCA basis in our model; see App. F. However, we found that this basis contained small scale variation, even in the leading components, making it ineffective at modeling the fixed effect.

---

> > ### Comment · Reviewer_UDtq · 2024-08-07
> >
> > I am satisfied by the rebuttal and have upgraded my decision to Accept. I suggest summarizing the key clarifications in the camera-ready draft if the final decision is indeed an Accept.

---

> > > ### Author Response · Authors · 2024-08-08
> > >
> > > Thank you again for providing constructive comments during the review period and for considering our rebuttal. If accepted, we will make sure to provide the necessary clarifications in the camera-ready version of the manuscript.

---

### Official Review · Reviewer_jjxT · 2024-07-09

**Soundness:** 3
**Presentation:** 2
**Contribution:** 2
**Rating:** 5
**Confidence:** 1

**Summary:**

The paper considers uncertainty quantification for one-dimensional regression tasks, where the observations are noisy. A rather advanced additive model considering invariances under space-time unitary transformations is proposted. Numerical experiments demonstrate the superiority of the approach over other state-of-the-art methods.

**Strengths:**

- Due to the strong modeling assumptions, on synthethic datasets, the method can recover the true signal even in situations where there is lots of noise and classical methods (such as GPs) may fail.
- The methods seems to be useful to analyze the Berkley growth spurt data set.

**Weaknesses:**

- It is unclear whether the work will be of interest to the wider machine learning audience and the considered topic and datasets feel more of a niche.
- Despite the well-written introduction, the paper was quite difficult to understand due to the advanced math and notations -- after reading it I am still quite unsure why I should really care about functional mixed models and their Bayesian inference.

**Questions:**

I am slightly puzzled as of why NeurIPS was chosen as a venue for this paper - the work may not get the appreciation here it deserves. I spent quite some time on trying to understand the paper, but its motivations and applications to wider machine learning remain unclear. Perhaps the paper may be better suited towards a statistics journal, such as the ones referenced often in the paper?   If more general motivations are clarified, I am considering to change my score, but accessibility issues are remaining.

**Limitations:**

All limitations are adequately addressed.

---

> ### Author Rebuttal · Authors · 2024-08-05
>
> We appreciate the reviewer's concern regarding accessibility to the broader machine learning community. If the manuscript is accepted, we plan to simplify notation as much as possible and provide more intuitive descriptions of some of the mathematical concepts.
>
> There were several factors that motivated us to choose NeurIPS as the outlet for this work.
>
> **1.** Functional data is arising as a common object in various applications, including computer vision and biomedical imaging, and functional data models are becoming increasingly important in machine learning, see e.g., Rao et al., Modern Non-linear Function-on-function Regression, 2023 and Rao et al., Nonlinear Functional Modeling Using Neural Networks, 2023. Furthermore, employing mixed models with random effects to better model correlated input data in neural networks is fast gaining traction, see e.g., Simchon et al., Integrating Random Effects in Deep Neural Networks, 2023.
>
> **2.** The field of shape analysis and more broadly geometric data analysis on quotient spaces, which provides part of the motivation behind the proposed modeling framework, falls under the umbrella of machine learning with broad applications in computer vision, graphics and biomedical imaging.
>
> **3.** In both, functional data analysis and shape analysis, geometry and invariance to nuisance variation play an important role and aid in model formulation and estimation. Indeed, these ideas can be applied more broadly in machine learning and there is increasing interest in the use of geometry and invariance in neural networks; see Bronstein et al., Geometric Deep Learning: Grids, Groups, Graphs, Geodesics, and Gauges, 2021 and references within.
>
> **4.** Our motivation stems from understanding the type of signal that can be recovered by a neural network or any probabilistic model from complex input data in the presence of nontrivial symmetries and geometric information. We have considered one specific instance of this, versions of which have been extensively studied in the statistics community.

---

### Official Review · Reviewer_JKM1 · 2024-07-11

**Soundness:** 3
**Presentation:** 3
**Contribution:** 2
**Rating:** 5
**Confidence:** 2

**Summary:**

For the problem of the reliable recovery of a fixed effect function $\mu$, this paper focuses on sampling from and summarizing the posterior distribution of a fixed effect function $\mu$ in a functional mixed model with random object-level phase and amplitude components, without a finite-rank covariance assumption on the error process.

----
Post-rebuttal: The rebuttal addressed most of my concerns and I would like to raise my score.

**Strengths:**

1）	This paper tries to solve a challenging task, recovering the size-and-shape of an unknown function by sampling.
2）	The theoretical analysis is interesting, e.g., Derivation of Metropolis–Hastings acceptance ratio.
3）	The authors also give the convergence analysis by numerical experiments.

**Weaknesses:**

1)	It seems that we should give different prior models for different functions.
2)	The parameter sensitivity of $\theta$ and the running time of the proposed method should be given.

**Questions:**

1)	Neural networks can also recover functions by sampling. What are the advantages of the proposed method compared to neural networks?
2)	How can we recover arbitrary by a fixed model in Eq.(3)?

---

> ### Author Rebuttal · Authors · 2024-08-05
>
> **Prior Models** We want to address this comment from two different perspectives. First, depending on the real data scenario, the choice of the type/number of basis functions in the model for the fixed effect function $\mu$ and the size-and-shape altering random effect $v_i$ can be different, e.g., for growth rate functions we used the modified Fourier basis to model the fixed effect $\mu$, while for the PQRST complexes we used B-splines. This choice is data dependent. At the same time, different types of basis functions can be used for the fixed effect function and the size-and-shape altering random effect. In general, we recommend the use of B-splines for the size-and-shape altering random effect since they have compact support and are able to capture local variation around the fixed effect function. In the end, the prior distributions are specified for the basis function coefficients.
>
> **Computational Cost and Sensitivity Analyses** Lines 366-370 provide information about the computational cost of MCMC sampling for the proposed Bayesian model. In short, we require ~111 minutes to yield 100,000 posterior samples for inference. The figure in the attached file contains sensitivity analyses with respect to (i) the number of basis functions used to model the fixed effect $\mu$, (ii) the number of basis functions used to model the size-and-shape altering random effect $v_i$, and (iii) the value of the concentration hyperparameter $\theta_\gamma$ in Prior Model 2 for the phase functions (size-and-shape preserving random effect). Through these experiments, we show that posterior inference is robust in almost all scenarios, except when the number of basis functions to model the fixed effect is under-specified (this is not unexpected since the ground truth $\mu$ does not lie in the span of the basis functions used in the model).
>
> **Neural Networks** We agree that neural networks can be used to approximate posterior distributions and their functionals; in other cases, they can aid in designing more efficient MCMC samplers by optimizing the proposal distribution such as in Li et al., A Neural Network MCMC Sampler That Maximizes Proposal Entropy, 2021. We have not explored these directions and leave them for future work. The primary goal of the current paper is to introduce a Bayesian functional mixed model, motivated by size-and-shape models in the context of shape analysis, that accounts for size-and-shape altering as well as size-and-shape preserving random effects. The novelty is in treating phase variation in functional data as a size-and-shape preserving random effect. We use standard MCMC for posterior inference, but plan to consider other approaches for approximating the posterior distribution under this Bayesian model in the future.
>
> **Generative Model (3)** In (3), we specify a generative model for functional data that incorporates a fixed effect function $\mu$, a noise process $\epsilon_i$, a size-and-shape altering random effect $v_i$, and a size-and-shape preserving random effect $\gamma_i$. The ability to simulate arbitrary functional data based on the model in (3) depends on the assumptions, e.g., smoothness, etc., on each of these model components. That said, this is a very general model, and to the best of our knowledge, the first one to consider both, size-and-shape altering and size-and-shape preserving random effects.

---

### Official Review · Reviewer_qjNQ · 2024-07-15

**Soundness:** 3
**Presentation:** 2
**Contribution:** 3
**Rating:** 7
**Confidence:** 4

**Summary:**

The paper studies the problem of recovering a fixed effect function µ in functional mixed models, where measurement errors and object-level phase variations make the task difficult. It focuses on disentangling the size-and-shape characteristics of µ, which remain invariant under certain transformations. The authors hypothesize that it is feasible to reliably recover the size-and-shape of µ using a Bayesian functional mixed model framework. The formulation is justified by numerical experiments on synthetic and real data.

**Strengths:**

1. The presentation over all is clear, self-contained, and precise. While this might sound standard in the past, I find these properties rare in papers these days, so I appreciate them even more.
2. The formulation is carefully backed up by justifications for the choice of components (e.g., basis functions, prior distributions) as well as numerical experiments.

**Weaknesses:**

## Significance:



1. The main proposal of this paper is to integrate phase variations into the model in a phase-preserving manner (equations 3 and 4). Nonetheless, I am struggling to understand what is the significance of such a model. More precisely,
    1. Is this a reasonable model for real applications? Namely, are there some practical applications in which we have reasons to believe the phase functions act on the fixed effect mu in a phase-preserving way?
    2. What are the main conceptual benefits and drawbacks of phase-preserving compared to value-preserving actions? The only answer I seem to find is in lines 142-147, but I do not understand the statement there as I detail in “Clarity”.
2. The comparison of the proposed method with warpMix on real data (Figure 4) seems limited, and this comment relate to the one above on real applications. It is hard to say which method is better when there is no ground-truth - the two methods give different estimates of mu, but it is hard to say which one makes more sense.

## Clarity:



1. Lines 142-147: Why is “The equivalence class of functions having the same size-and-shape under the norm-preserving action is in an appropriate sense ‘larger’ than the one under the value-preserving action”?
    1. The explanation following the colon is not sufficient, since the same could be said for value-preserving action: for any g and gamma(t), one can find f(t) := g(gamma^{-1}(t)) such that f(gamma(t)) = g(gamma^{-1}(gamma(t))) = g(t).
    2. The explanation in footnote 1 is a bit hand-wavy and it would be great if rigorous proof could be provided or referred to.

## Novelty:


The idea that the parameterization of the model is not minimal and therefore there is the ambiguity of having an equivalence class of solutions has been there for a long time, as the authors also point out in equation (2). Since (2) is an example in geometric vision which i am a bit more familiar with, I would like to point out more high-level connections here.


1. A basic idea to resolve the ambiguity is to enforce constraints that favors certain elements in each equivalence class than others. This is more or less what the authors are doing near line 76. Note however that: while two sets of parameters can be equivalent in the sense that if there were no noise then the two sets of parameters explain the observation equally well, in the noisy case the two sets of parameters could handle noise very differently. Therefore, one needs to be smart on which elements in each equivalence class to pick. For example, the work of [A] argues that picking a particular transformation (normalization) in the parameters improves numerical stability of the estimation in the noisy case.
2. Following the same line, two sets of “equivalent” parameters could also be favored differently when the data are corrupted by outliers. An alternative idea to resolve the ambiguity is to estimate the entire equivalent class (rather than finding an element of it), which is for example the path explored in the work of [B].

So far i do not see a strong need for the current paper to exhaust all options, but it would be great if the authors could discuss on how optimal the choice of line 76 is, what might be potential alternatives, etc.

[A] R. I. Hartley, “In defense of the eight-point algorithm,” IEEE Transactions on Pattern Analysis and Machine Intelligence, 1997.

[B] Ding, T., Yang, Y., Zhu, Z., Robinson, D. P., Vidal, R., Kneip, L., & Tsakiris, M. C, “Robust homography estimation via dual principal component pursuit”, In the IEEE/CVF Conference on Computer Vision and Pattern Recognition, 2020.

**Questions:**

## Minor comments / questions:



1. Could you comment on how does using MCMC to get parameters from the posterior compare to using optimization, e.g., by writing down the log likelihood and then maximize it over the parameters?
2. The introduction was somewhat confusing, as illustrated below.
    3. The first paragraph motivates the reader with the Berkeley growth data example, saying that a problem for modeling is that the sample size is smaller than the dimension observation, so one must assume prior / constraints in the hypothesis class. Yet the model in equation (1) is rather general and does not assume much prior for recovery.
    4. Now the second paragraph makes the model even more complicated and harder to estimate by introducing phase variability. In particular, why isn’t model (1) sufficient and why does phase variability make sense?

    To sum up, I think there are multiple problems motivated and highlighted along the way which somehow buries the key issue of phase variability.

3. The reviewer is grateful that some basics on phase functions and size-and-shape-preserving transformations are provided in section 2. Nonetheless, when I read Section 2 (before going through the rest of the paper) it is unclear why are these properties important - are they going to contribute to the solution? Which key properties allow you to do something that previously was not done? This for example can be said in the preamble of section 2 to make the flow smoother.
4. Line 119: gamma in Gamma?

**Limitations:**

See above

---

> ### Author Rebuttal · Authors · 2024-08-06
>
> **Significance, Size-and-Shape Preserving Transformations** Broadly, for *any* application involving functional data, our model has a two-fold motivation: (i) norm-preserving action $D_\gamma$ on fixed effect $\mu$ and size-and-shape preserving random effect $v_i$, expressed as linear combinations of orthonormal bases, allows us to rotate the basis systems to better align with observed data, and (ii) isometric group actions play a key role in finite and infinite-dimensional shape analysis to account for nuisance variation (Srivastava et al., Functional and Shape Data Analysis, 2016). For (i), modeling flexibility enabled by $D_\gamma$ results in more parsimonious models, especially since $\mu$ is represented via a finite, and potentially low-dimensional, linear combination of basis functions. This has applied implications since the type of basis is chosen apriori. The viewpoint in (ii) allows us to formulate models and carry-out computations in a geometric fashion.
>
> **Comparisons** Quantitative comparisons to warpMix on synthetic data generated via the warpMix model in Sec. 4.1 show that our model performs better in terms of recovery of $\mu$. In Sec. 4.2, we consider two real datasets: growth rate functions and PQRST complexes. In the first, warpMix only recovers a single growth spurt while our model recovers two growth spurts, a smaller initial one and later pubertal one. In the second, warpMix fails to produce an estimate. We provide results for more real datasets in App. H. In general, warpMix oversmooths the estimate of $\mu$ resulting in fewer geometric features than expected. Finally, posterior MCMC samples from our model can be used to quantify uncertainty, which is an important applied consideration (warpMix provides a point estimate).
>
> **Equivalence Classes** The example you provide demonstrates surjectivity of the operator from a value-preserving action, but such an operator is not unitary since it does not preserve the $\mathbb L^2$ norm. Operator $D_\gamma$ on the other hand is unitary for every $\gamma \in \Gamma$. Let $[f]_n:=$\{$f(\gamma)\sqrt{\dot \gamma},\gamma\in\Gamma$\} be the equivalence class of a function $f$ under the norm-preserving action and $[f]_v:=$\{$f(\gamma),\gamma\in\Gamma$\} be the equivalence class under the value-preserving one. Class $[f]_v$ contains functions that have the *same* sequence of ordinate values as $f$ - image $t \mapsto f(t)$ is preserved. On the other hand, $[f]_n$ contains functions with new ordinate values (due to scaling factor $\sqrt{\dot \gamma}$) that are originally not in $f$, and new extrema may be created. Intuitively, thus, $[f]_n$ is "larger" than $[f]_v$. A rigorous proof that compares sizes of two open sets (e.g., open balls) in two norm topologies induced by two quotient metrics (w.r.t. to the equivalence relation) can be provided. However, the following simple example with Brownian motion $W(t)$ illustrates the above point. Let $X(t)=W(c t)$ with $c>0$ be a time-changed process, corresponding to a value-preserving action by scaling of time on some interval in $(0,\infty)$. Then, the law of $X$ is singular w.r.t. the law of $W$ for every value of $c$ except $c=1$. However, under the norm-preserving map, $X(t)=W(\gamma(t))\sqrt{\dot \gamma(t)}$ is absolutely continuous w.r.t. $W(t)$ for any $\gamma \in \Gamma$ (Example 6.5.2 in Bogachev, Gaussian Measures, 1998).
>
> **Equivalent Solutions** Indeed, line 276 introduces a post-hoc constraint to choose an appropriate element of the equivalence class of $\mu$ in a data-driven manner, ensuring that the resulting $\mu$ is associated with identity phase $\gamma_{id}(t)=t$; this step is necessary since functional data comes under arbitrary phase variation. This is commonly referred to as orbit centering where the average of phase functions is used to choose an orbit representative (Chapter 4 of Srivastava et al., Functional and Shape Data Analysis, 2016). *This choice is optimal with respect to minimizing the extrinsic Frechet variance of the $\gamma_i$s.* An alternative that can be used in the presence of phase outliers is the Frechet median. We thank the reviewer for pointing us to the two references. For [A], we agree that choosing an appropriate equivalence class representative is difficult in the presence of varying levels of noise. In our model, we use finite-dimensional priors on phase, which help us constrain the size of equivalence classes. For [B], it is an interesting idea to recover the entire equivalence class instead of a representative. However, the parameter spaces for our model are (high or) infinite-dimensional (even though some dimension reduction is enforced via prior formulations). Under the one-dimensional Prior Model 1 on phase, one can potentially explore the entire equivalence class of the posterior mean of $\mu$.
>
> **Frequentist Inference** Maximum likelihood estimation of parameters would yield the maximum aposteriori estimate corresponding to our model under appropriate non-informative prior choices. Despite higher computational cost, we favor Bayesian inference since it allows us to easily quantify posterior uncertainty.
>
> **Introduction** We agree that, while important, the main issue motivating our framework is not the data dimension. Instead, the motivation stems from the fact that children can undergo different numbers of growth spurts of different magnitudes at different times. This implies that the data contains phase variation, which is not accounted for in model (1) (an implicit assumption in (1) is that there is no time variation in growth spurts). Phase variation in our model acts as a size-and-shape preserving random effect that allows better alignment of the magnitude and timing of growth spurts across observations. If accepted, we will clarify our motivation. Methodological motivation of our model, as opposed to one such as warpMix with value-preserving phase variation, is given in lines 31-45.
>
> We will use $\gamma\in\Gamma$ in line 119.

---

> > ### Comment · Reviewer_qjNQ · 2024-08-11
> > **Thank you for your rebuttal**
> >
> > Thanks to the reviewers for the response! I am clear on all except the following concerns:
> >
> > **Clarity 1.1**, **Significance 1.2** / **Equivalence Classes**: I am unsure how this intuition helps conclude that $[f]_n$ is larger than $[f]_v$. From your argument, I can agree that functions in $[f]_v$ satisfy the property you mentioned (i.e. same sequence of ordinate values), and that functions in $[f]_n$ do not satisfy that property. However, functions in $[f]_n$ could apriori satisfy some other properties that functions in $[f]_v$ do not satisfy, making $[f]_n$ small.
> > - I do not have any clue if that is indeed the case, but your argument does not prevent such cases from happening (if I'm understanding it correctly), which is why I could not understand the intuition. Do I miss something here?
> >
> > **Significance 1.1**: I guess I can agree on what (i) and (ii) do, i.e., what norm-preserving action and isometric group actions do, but that's not what I was trying to ask. Maybe I am not understanding your response. My original questions are regarding concrete applications, in which we know the better way to model the signals is by norm-preserving actions rather than the alternatives (e.g., value-preserving actions or area-preserving actions)
> > - Are there such applications? Why are norm-preserving actions a better model (other than empirical numbers showing its better)?
> > - Maybe you already had them in the paper/response: are you saying that equation (2) is an example? Or time warping?

---

> > > ### Author Response · Authors · 2024-08-12
> > >
> > > Thank you for your additional questions.
> > >
> > > **Clarity 1.1, Significance 1.2 / Equivalence Classes**: You are indeed right that $[f]_n$ need not be larger than $[f]_v$ for every function $f$ - that necessarily depends on the function $f$. Our claim on sizes of equivalence classes, however, is with respect to the measure induced on the quotient $\mathbb{L}^2/\sim_n$ compared to the quotient $\mathbb{L}^2/\sim_v$ starting from a non-degenerate measure on $\mathbb{L}^2$; here $\sim_n$ is the equivalence relation on $\mathbb{L}^2$ induced by the norm-preserving action, and similarly $\sim_v$ for the value-preserving action. The example in our response illustrates this when the measure on $\mathbb{L}^2$ is the Wiener measure (with support on continuous functions): its push forward under $\sim_v$ is effectively singular with respect to the Wiener measure, and thus prescribes zero mass to $[f]_v$; in contrast, the push forward of the Wiener measure under the quotient map corresponding to $\sim_n$ is absolutely continuous with respect to the Wiener measure, and prescribes positive mass to $[f]_n$ for every $f$, since the norm-preserving action is an isometry of $\mathbb{L}^2$. Indeed, this does not preclude the possibility of constructing a measure on $\mathbb{L}^2$ that violates the preceding claim; but, in our opinion, such measures are not “natural” since they are not compatible with isometries of $\mathbb{L}^2$, which in this case is the norm-preserving action.
> > >
> > > Further, ignoring the random effect, if the fixed effect function is mis-specified with fewer basis functions than required to capture the correct number of extrema, the value-preserving action cannot rectify this, whereas the norm-preserving one can. This has modeling significance since, in a way, the norm-preserving action makes the model more robust. Of course, the size-and-shape altering random effect can introduce new extrema, but only on the individual level and not the population level.
> > >
> > > The discussion has brought to light the need to be clearer with the claim on sizes of the equivalence classes in the paper, and we thank you for this.
> > >
> > > **Significance 1.1**:
> > >
> > > 1. Using the Hilbert space $\mathbb L^2([0,1],\mathbb{R})$ as the representation space for the observed functional data is popular, mainly due to availability of an orthonormal basis that enables working with the basis coefficients and facilitates dimension reduction. Given this, modeling the presence of phase variability via the $\mathbb L^2$-norm preserving action of $\gamma$ allows us to use the geometry of the quotient space to develop computational tools; the value-preserving action, in contrast, preserves the uniform $\mathbb L^\infty$-norm $\sup_t|f(t)|$. Moreover, an action preserving any $\mathbb L^p$ norm for $1 \leq p <\infty$ can be converted to an action that preserves the $\mathbb L^2$-norm via the pointwise transformation $f(t)\mapsto \text{sgn}(f(t))|f(t)|^{p/2}$ for every $t \in [0,1]$. The norm-preserving action allows more flexibility in modeling the fixed effect function $\mu$ as compared to the value-preserving action since time warping is accompanied by local rescaling. While a definitive answer to your question is difficult, in many applications including biology, medicine and biometrics, the size-and-shape of $\mu$ is of primary interest for inference. One application that motivates our model, as opposed to one that utilizes a norm-preserving action such as warpMix, is in recovery of the underlying number and magnitude of growth spurts based on a sample of growth rate curves (from the Berkeley growth data). It is well known in this context that children undergo different numbers of growth spurts with different magnitudes occurring at different times. This implies that the data contains phase variation, and the primary task of interest is recovery of the size-and-shape (magnitude and number) of the population level fixed effect function $\mu$. Our estimate, which contains two growth spurts, a smaller initial one and a larger pubertal one later on, agrees with expectations, since it has been shown that children undergo more than one growth spurt during the growth process. The warpMix estimate on the other hand only contains only a single growth spurt.
> > >
> > > 2. It is commonplace in geometric statistics and statistical shape theory to incorporate the action of a nuisance transformation as an isometric group action (see, e.g., the books Shape and Shape Theory by Kendall et al. and Functional and Shape Data Analysis by Srivastava et al.). For example, when size-and-shape of a landmark configuration is of interest, since rotations and translations do not change the configuration's size-and-shape, they are nuisance, and due to use of the Euclidean norm on the landmarks, act isometrically on the configuration. Model (2) in our paper uses this in its formulation, and inspires our model for functional data in the presence of phase variation.

---

### Official Review · Reviewer_yD41 · 2024-07-28

**Soundness:** 3
**Presentation:** 3
**Contribution:** 3
**Rating:** 5
**Confidence:** 3

**Summary:**

In this paper is proposed a mixed model in a functional Hilbert space for a size-and-shape (a type of geometric property) of a square-integrable fixed effect. To this end, the authors consider an isometric action of the infinite-dimensional group of phase functions. Synthetic and real experiments and comparisons with respect to another method in literature are reported, demonstrating how a posterior mean, and under the proposed model, captures the main properties of the function.

**Strengths:**

+ A Bayesian functional mixed effects model with an unrestricted form of phase variation is presented.

+ A novel perspective of inferring a size-and-shape function, a type of geometric constraint.

+ In general, the paper is concise and direct. Most explanations and discussions are provided. It is easy to follow, up to some points in formulation.

**Weaknesses:**

- I think the notation could be improved, representing, for example, the use of vectors, matrices and scalars differently. In some cases, including dimensionality may help readers. Moreover, some symbols are different along text (see the transpose operator).

- A couple of linear combinations of basis functions are assumed. The rank of every subspace is Bf and Br. This type of formulation has been proposed in previous approaches, following an LDA style. The rank of every subspace is known a prior.

- The authors could include a practical motivation of the study, where they could apply their approach and why it is important.

**Questions:**

How is the rank of both subspaces fixed? By hand? How sensitive is the solution with respect to these parameters? Could Bf and Br be inferred automatically? This seems to be a key factor in the model, but the authors do not provide any discussion in this line.

The authors propose to use a B-spline basis, so a pre-defined one. Why this type of function? Note that we can find many piecewise polynomials in literature.

Could both basis functions be learned jointly with weight coefficients?

In real data, n=93, but when the authors use synthetic one, n=100,000. I cannot understand why the difference needs to be so big. In my opinion, if the goal is to sort out the problem in scenarios where the amount of data is limited, this should be also considered in the synthetic scenario.

Figure 1 (b) and (c) are not cited in the introduction.

The method consistently outperforms warpMix on synthetic data. While I like this comparison, I think the authors should consider more datasets for this evaluation. In addition to that, what about computational cost? I would like to see the trade-off accuracy vs. cost for both methods.

What about failure cases?

**Limitations:**

The authors do not analyze properly the limitations of the paper. In any case, I do not think the work in this paper can produce a negative social impact.

---

> ### Author Rebuttal · Authors · 2024-08-05
>
> **Notation, Figure 1, Sample Sizes** We agree that the notation is dense in certain sections. If accepted, we will try to simplify notation. We denote vectors and matrices in bold and functions and scalars in regular font. We will include dimensionality when appropriate. Figure 1 (b-c) shows the effect of the norm-preserving action $D_{\gamma}$ on $f$ (referred to on line 26 in Section 2). These images are provided in Figure 1 due to the page limit. In all simulations, we use $n=30$ (line 289). Thus, sample sizes in simulations and real data examples are comparable. The $N = 100,000$ is the number of posterior samples that are used for inference (line 280).
>
> **Motivation** Incorporating random effects in neural networks is a popular technique to account for correlations in input data (e.g., Simchoni et al., Integrating Random Effects in Deep Neural Networks, 2023). In this context, our main motivation stems from ascertaining what may be reliably recovered by a Bayesian model when inputs are functions, and the objective is to estimate a population-level fixed effect function $\mu$ in the presence of two different types of random effects: size-and-shape altering one ($v_i$) and size-and-shape preserving one ($\gamma_i$, phase variation). Simulated and real data examples show that our model better recovers geometric features of $\mu$ than the SOTA model warpMix. We expect the understanding gained via the simple Bayesian model to have important consequences when overparameterized neural networks are used with input functional data. We discuss and analyze two motivating datasets in Sec. 1 and 4.2. The proposed framework is general with many applications; results for other real datasets are in App. H.
>
> **Type of Basis and Joint Estimation** Our model allows use of *any* basis for the fixed effect $\mu$ and the size-and-shape altering random effect $v_i$. *We clarify that, despite choosing the type of basis apriori, the norm-preserving action $D_{\gamma}$ rotates the basis system toward the data, allowing us to learn a data-driven basis for $\mu$. Thus, theoretically, the apriori choice of basis is unimportant for recovery of size-and-shape of $\mu$, a crucial implication of our geometric formulation of the problem.* Below, we provide practical guidance on this issue.
>
> Orthonormality: The basis should be orthonormal since our model is motivated by viewing the norm-preserving action as a rotation of the basis system.
>
> Data complexity, fixed effect: Choice of basis for $\mu$ should be motivated by the structure of given data, e.g., we use a Fourier basis for growth rate data, but B-spline basis for PQRST complexes; this is based on prior belief of whether $\mu$ contains periodicity (growth rate data) or sharp geometric features (PQRST complexes). We found that a functional PCA (FPCA) basis, estimated using the data, is ineffective in modeling $\mu$ as it captures noise and small-scale variation (App. F).
>
> Random effect: We prefer a basis with compact support, e.g., B-splines, to capture local variation and finer features as compared to the fixed effect.
>
> We appreciate the comment on joint estimation, which can be quite challenging: one needs to disentangle MCMC errors related to estimating the functional subspace (what is the optimal basis and how many?) from that of estimating the size-and-shape of $\mu$. We will pursue this in the future for the fixed effect, which is the primary object of interest for inference, in two ways: by (i) placing a prior on the Stiefel manifold of orthonormal frames, or (ii) using ideas from Matuk et al., Bayesian Modeling of Nearly Mutually Orthogonal Processes, 2023.
>
> **Number of Basis Functions** This is chosen apriori, and can be informed by exploratory analyses, e.g., FPCA. We found that posterior inference is robust to (i) under- and over-specification of the number of basis functions for the size-and-shape altering random effect, and (ii) over-specification for the fixed effect. It is somewhat robust to under-specification for the fixed effect; see figure in the attached file. Thus, one may choose a large basis for the fixed and random effects. Alternatively, we could treat $B_f$ and $B_r$ as random and infer them. We leave this for the future since posterior inference becomes more complex: we may have to use Reversible Jump MCMC to allow the parameter space to change dimension.
>
> **warpMix Comparisons** We carefully chose simulations to ensure fair comparisons to warpMix. For quantitative evaluation, we use the criterion used in their work (line 318). Simulated Example 2 is particularly illuminating wherein the datasets are generated using the warpMix model, which uses a value-preserving action on $\mu$ and $v_i$ rather than a norm-preserving one. Our model qualitatively/quantitatively outperforms warpMix in all considered simulations. We provide comparisons to warpMix on real data in Sec. 4.2 and App. H; our model recovers a fixed effect with more pronounced geometric features. Also, since we use a Bayesian model, we are able to quantify uncertainty via 95% credible intervals; this is not straightforward in warpMix (result is only a point estimate).
>
> Computationally, warpMix is more efficient: we rely on MCMC while warpMix uses iterative likelihood optimization. We discuss computational cost of our method in Section 5, lines 366-370 and mention that there is room for improvement in MCMC efficiency (most efficient MCMC was not our primary focus). For comparison, 10 iterations in warpMix take ~5 minutes as compared to ~111 minutes needed by MCMC for 100,000 posterior samples for inference.
>
> **Failure Cases/Limitations** As mentioned earlier, we cannot reliably recover the true fixed effect when the number of basis functions is under-specified. This is not unexpected since the ground truth $\mu$ does not lie in the span of the basis functions used in the model. We discuss limitations in Section 5, including lack of theoretical support and computational cost.

---

> > ### Comment · Reviewer_yD41 · 2024-08-13
> >
> > Thank you for the clarifications!
> >
> > I am happy with the answers that have been provided. In any case, although recovering the rank is a complex problem, this point should not be hidden and must be included as part of the future work.

---

> > > ### Author Response · Authors · 2024-08-13
> > >
> > > Thank you for considering our rebuttal. If accepted, we will make sure to list future work related to rank/joint estimation as future work.

---

### Author Rebuttal · Authors · 2024-08-06

We thank all reviewers for their careful consideration of our manuscript and constructive comments.

**Significance and Motivation** Employing mixed models with random effects to better model correlated input data in neural networks is fast gaining traction (e.g., Simchon et al., Integrating Random Effects in Deep Neural Networks, 2023). In this context, when the inputs are functional data observed with *arbitrary* phase variability, it is important to understand what feature/property of the fixed effect function $\mu$ may be reliably recovered. Our main contribution is the proposal, and investigation, of a geometric reformulation of the problem: we use the phase component in the probabilistic model as a space-time unitary transformation, and then determine a data-driven optimal rotation of the model's coordinates to recover the size-and-shape of $\mu$, with corresponding uncertainty estimates. The property of the norm-preserving action being an infinite-dimensional rotation of $\mathbb{L}^2$ is key in the formulation of the proposed model; this action is an isometry under the $\mathbb{L}^2$ metric. Since the fixed effect function (and the size-and-shape altering random effect) are eventually represented using a finite-dimensional basis set, such rotations allow us to align the $\mathbb{L}^2$ coordinate system of model components to the coordinate system of each observation, thus resulting in a more parsimonious model. Quantitative and qualitative evaluations on simulated and real datasets confirm these claims.

**Implementation** Our main contribution is a novel geometric perspective on the problem of reliably recovering the size-and-shape of a fixed effect function $\mu$ using a Bayesian functional mixed model with *unconstrained* (infinite-dimensional) size-and-shape altering and preserving random effects, in the presence of noise. As such, the computational algorithm used for posterior inference via MCMC serves more as a proof of concept rather than a definitive (optimal) computational tool, but nevertheless outperforms the current state-of-the-art (SOTA). There are many alternative algorithms that can be used to approximate the posterior distribution including variational approaches, and neural networks for intractable likelihoods and MCMC (e.g., Li et al., A Neural Network MCMC Sampler That Maximizes Proposal Entropy, 2021).

**Additional Results in Attached File** The attached PDF file contains a figure and a table. In the figure, we show results of more extensive sensitivity analyses to misspecification of three hyperparameters: $B_f$ (number of basis functions used to model the fixed effect function $\mu$), $B_r$ (number of basis functions used to model the size-and-shape altering random effect $v_i$) and $\theta_\gamma$ (concentration hyperparameter in Prior Model 2 (PM 2) on the size-and-shape preserving random effect or phase function $\gamma_i$). Each plot shows the centered estimate of the posterior mean (red), 95% credible interval (dashed blue) and ground truth $\mu$ (black). Rows 1-3 show results for under-specified, correctly specified and over-specified values of hyperparameters, respectively. Posterior inference, in terms of the posterior mean for $\mu$ and its uncertainty as ascertained via the 95% credible interval, is very robust to (i) over-specification of $B_f$, (ii) under or over-specification of $B_r$, and (iii) under or over-specification of $\theta_\gamma$. When $B_f$ is under-specified, we are unable to reliably recover the ground truth $\mu$ since it does not lie in the subspace spanned by the specified basis functions. Thus, in general, we recommend specifying a larger number of basis functions to model the fixed and size-and-shape altering random effects.
The table reports a more comprehensive quantitative evaluation, in terms of estimation accuracy for the fixed effect function $\mu$, on five simulated datasets. Rows 1-2 consider data simulated from our model under Prior Model 1 (PM 1) and PM 2 for phase functions. Rows 3-5 consider data simulated using the warpMix model with default parameter values. We compare estimation results produced using our model (columns Model 1-F through Model 2-B where the number refers to the prior model on phase and the letters F or B refer to the Fourier or B-spline bases) to those produced using warpMix (Claeskens et al., Nonlinear Mixed Effects Modeling and Warping for Functional Data Using B-splines, 2021) and the mean+noise model proposed in Cheng et al., Bayesian Registration of Functions and Curves, 2016 (labeled as BRFC). As seen in the table, our model outperforms warpMix and BRFC in all of these simulation scenarios.

---

### Author Response · Authors · 2024-08-12

Dear Reviewers,

With the discussion period coming to an end tomorrow, we wanted to thank you again for providing constructive comments and considering our rebuttal. We are happy to answer any remaining questions.

---

### Decision · Program_Chairs · 2024-09-25

**Decision:**

Accept (poster)

**Comment:**

The reviewers found that the paper introduces an interesting infinite-dimensional Bayesian mixed model, offering a new perspective on incorporating geometric constraints. They acknowledged the efforts made during the rebuttal, which provided more convincing results. However, they raised concerns about the clarity of the writing and suggested several additions to better motivate the exposition of the method. These revisions should be incorporated into the final version to enhance the quality of the manuscript. I recommend acceptance of this paper.